environmental science/behaviour/ecosystems

animal welfare, broiler chickens, land use, environmental policy, dietary consumption

**Author for correspondence:**
Iris Chan
e-mail: irischan@nyu.edu

# The 'sustainability gap' of US broiler chicken production: trade-offs between welfare, land use and consumption

## Iris Chan, Becca Franks and Matthew N. Hayek

Department of Environmental Studies, New York University, New York, NY 10012, USA

 IC, 0000-0002-4380-2164; BF, 0000-0002-7558-6718; MNH, 0000-0001-9792-4362

In 2018, over nine billion chickens were slaughtered in the United States. As the demand for chickens increases, so too have concerns regarding the welfare of the chickens in these systems and the damage such practices cause to the surrounding ecosystems. To address welfare concerns, there is large-scale interest in raising chickens on pasture and switching to slower-growing, higher-welfare breeds as soon as 2024. We created a box model of US chicken demographics to characterize aggregate broiler chicken welfare and land-use consequences at the country scale for US shifts to slower-growing chickens, housing with outdoor access, and pasture management. The US produces roughly 20 million metric tons of chicken meat annually. Maintaining this level of consumption entirely with a slower-growing breed would require a 44.6%–86.8% larger population of chickens and a 19.2%–27.2% higher annual slaughter rate, relative to the current demographics of primarily 'Ross 308' chickens that are slaughtered at a rate of 9.25 billion per year. Generating this quantity of slower-growing breeds in conventional concentrated animal feeding operations (CAFO) would require 90 582–98 687 km$^2$, an increase of 19.9–30.6% over the 75 577 km$^2$ of land used for current production of Ross 308. Housing slower-growing breeds on pasture, the more individually welfare-friendly option, would require 108 642–121 019 km$^2$, a 43.8–60.1% increase over current land use. Allowing slower-growing breeds occasional outdoor access is an intermediate approach that would require 90 691–98 811 km$^2$, an increase of 20–30.7% of the current land use, a very minor increase of land relative to managing slower-growing breeds in CAFOs. In sum, without a drastic reduction in consumption, switching to alternative breeds will lead to a substantial increase in the number of individuals killed each year, an untenable increase in land use, and a possible decrease in aggregate chicken welfare at the country-level scale. Pasture-based management requires substantial additional land use. These

results demonstrate constraints and trade-offs in animal welfare, environmental conservation and food animal consumption, while highlighting opportunities for policies to mitigate impacts in an integrated manner using a One Health approach.

# 1. Introduction

In 2018, the conventional broiler chicken production generated over 31.7 billion US dollars, with over nine billion chickens slaughtered in the United States [1], and the overall consumption of chicken meat in the USA continues to rise [2]. Broiler chickens are raised for meat, and are different breeds to chickens who lay eggs. As the demand for chickens increases, concerns also arise regarding the individual welfare of the chickens in these systems and the damage such practices cause to the surrounding ecosystems and environment, such as greenhouse gas emissions, freshwater pollution, and encroachment of feed cropland into native ecosystems [3]. Animal welfare concerns of chickens raised for meat in conventional concentrated feeding animal operations (CAFOs) include contact dermatitis [4], leg lameness [5–7], ascites [8], locomotor problems [9] and subtherapeutic antibiotic dosage [10]. Consequences of such intensive fattening systems include physical and emotional stress from crowded spaces [11] and poor litter quality [7,12], which increases microbial activity [13], amplifying the risk of disease transmission [14] and the proliferation of antibiotic-resistant bacteria [15]. Because the USA is virtually self-sufficient in chicken production, US consumption preferences greatly influence aggregate chicken welfare at the country-level scale.

Animal welfare organizations have suggested alternative ways to produce broiler chickens from the conventional, intensive system and have secured corporate commitments to shift production for 277 million chickens to alternative methods as soon as 2024 [16,17]. Specifically, recommended practices include raising intermediate-growing breeds (determined by Global Animal Partnership (GAP) and the Better Chicken Commitment) (e.g. Ranger Classic and Ranger Gold), slow-growing breeds (Rowan Ranger), and introducing more spacious management systems (e.g. outdoor access or pasture-raised) [18,19]. However, shifting breeds and management practices has the potential to increase the number of animals raised, fed and slaughtered, entailing novel concerns for welfare and extensive land-use requirements, relevant to environmental conservation, when considering such shifts at a country-level scale [20]. Previous analysis has examined trade-offs between welfare-enhancing interventions and greenhouse gases [21], integrated welfare in life-cycle assessments [22] and examined organic versus conventional performance across a suite of environmental metrics [23]. Because human, animal and environmental health are interrelated at large economic and spatial scales, experts have previously called for a 'One Health' approach, which in part represents and models such consequences of shifting food animal production in an integrated manner. In the case of broiler chickens, such an approach has not previously been adopted to understand interrelated welfare and land-use impacts of shifting broiler production at the country-level scale [24,25].

Producers and animal welfare advocates have expressed concern about the current fast-growing chicken breed, Ross 308 [26]. Because animal scientists primarily focus on efficiency of animal production, this highly efficient breed is presently used in all commercial-scale production in the USA. The Ross breed has been genetically selected to reach a market weight of roughly 2.93 kg in 47 days [27,28], primarily to produce large cuts of lean breast meat. However, this rapid growth can cause muscle strain, significant bone stress, and potential bone breakage [6]. These physical damages to chickens' bodies and their negative impacts upon their welfare could be lessened by raising broiler chicken breeds with slower-growth rates. For example, the slow-growing 'Rowan Ranger' (RR) chickens take an average of 69 days to reach a market weight of 2.43 kg. However, the additional time needed to raise these chickens is unsustainable. A third group of birds, the intermediate-growing breeds, which include Ranger Gold and Ranger Classic (RG and RC, respectively) may still allow mass production at a faster rate than the slow-growing breeds, and improved welfare than the fast-growing breeds. The breeds of chickens permitted by Global Animal Partnership (GAP) are in the Better Chicken Commitment (BCC) Policy [19]. While Ross chickens tend to have higher production than slower-growing chickens, slower-growing chickens have more leg meat, a better quality of meat, improved welfare in terms of activity (feather cover, gait score, breast cleanliness), more exploratory behaviour, better litter quality and lower mortality rates [5–7,9,29]. Positive animal welfare has also been linked to farmer well-being, further emphasizing the need to take a One Health approach to quantifying the multiple human, animal and environmental consequences of changing agricultural management [25].

Animal welfare commitment programmes to raise higher-welfare breeds entail uncertain consequences for aggregate welfare at the country-level scale. Shifting to slower-growing breeds that reach lower final slaughter weights would entail raising a greater population (total number of chickens in existence at a given time in the USA) and a higher slaughter rate (number of chickens killed per year). The population and the slaughter rate are both relevant for assessing aggregate welfare. Independent of annual slaughter rate (i.e. holding slaughter rate constant), higher populations of animals raised within fattening systems in which they face negative welfare issues throughout their lifetime aggregates to lower welfare overall compared with conditions that involve lower populations. Similarly, if a greater number of animals are slaughtered, more animals will endure inevitable stresses and harms of transportation and slaughter, reducing aggregate welfare (i.e. when considering the animal slaughter rate). The details of any aggregation procedure will greatly influence the final outcomes upon welfare, but without data regarding the number of birds affected under various production systems—reflecting both the population and slaughter rate—any such discussion of the relative trade-offs to welfare in the aggregate are impossible.

Considering and analysing aggregated welfare across large human populations and among countries has been embraced in human public health research for decades, through outcome metrics such as disability- or quality-adjusted life years. These methods of aggregating welfare have more recently come into favour as a method of analysing the welfare of animals. Recent analysis has examined consequences of shifting diets upon number of animal life years lost and/or suffered [30]. Theoretical concerns remain about how welfare should most appropriately be aggregated [31]. However, such recent research highlights that population size and slaughter rate are increasingly relevant prerequisite metrics for assessing the consequences of interventions upon animal welfare.

To reduce crowding and further improve welfare, broiler chicken producers can use alternative management systems to CAFOs. Currently, less than 1% of chickens are raised in a free-range or pasture-raised system [32]. According to the United States Department of Agriculture (USDA), free-range means 'producers must demonstrate to the Agency that the poultry has been allowed access to the outside' [33]. Pasture-raised systems, on the other hand, while not legally defined or certified by the USDA [33], are defined by third-party certifying organizations, like Global Animal Partnership (GAP) and Humane Farm Animal Care (HFAC). GAP and HFAC determine minimum space requirements for animal welfare standards for chickens raised for meat and certified pasture-raised birds [18,34]. Because of these increased space requirements per bird, more direct land is needed for more spacious management systems, potentially displacing wild spaces [3]. Chickens raised outdoors have higher activity rates and have better-reported quality and flavour of meat [35,36]. Outdoor access has shown few negative health consequences; however, raising chickens outdoors comes with increased risk of coccidia infection [37], for which vaccines are available, and predation [38]. The land-use consequences of outdoor access and pasture-raised management have not previously been quantified at scale.

Regardless of the management system used, shifting to slower-growing breeds, which have different nutrition requirements, would affect indirect land use via their unique nutrient requirements, hence the total quantity of crops required. Pastures and croplands for raising animals incur opportunity costs for production, potentially displacing forest habitats, ecosystem carbon [39] and biodiversity [40]. Total land-use requirements for poultry must account for both direct land use for the chickens themselves and indirect land use for their feed production.

Presently, GAP welfare commitments to slower-growing breeds and better management only encompass less than 300 million chickens annually [17] compared with the annual nine billion chickens slaughtered. Quantifying the potential aggregate welfare and land-use consequences of such shifts requires an understanding of the current populations, lifespan, weight gain and land use of US broilers. To simultaneously investigate potential trade-offs and optimize both aggregated welfare and land usage, animal scientists, environmental systems scientists and welfarists should work in tandem. These interdependencies highlight the need for more work adopting a One Health approach, that is, simultaneously considering animal welfare, environmental health, and human health and behaviour [24,25]. Using a demographic model that also considers land-use requirements, we ask the following questions: (1) *What size population of slower-growing chickens is required to produce the quantity of chicken meat currently consumed? (2) How do land use and feed requirements change for shifts toward slower-growing chicken production? (3) What are the land-use, feed, and population implications of shifts to outdoor access and pasture-raised chicken? (4) If current chicken land use was maintained but production shifted to slower-growth breeds and/or higher-welfare production methods, how much would consumption patterns have to change?* This study focuses on broiler chickens who are raised for meat, which are different breeds to chickens who lay eggs. The data analysed in this study is based on US production, but our analysis highlights potential consequences and trade-offs that warrant consideration in intensive production systems elsewhere.

# 2. Methods

## 2.1. Chicken demographics in the conventional system

To predict changes in broiler chicken demographics necessary for producing slower-growing breeds, we first created a demographic box model of the conventional chicken meat production system of Ross 308 raised in CAFOs. The US broiler chicken flock consists of a breeding flock to replace broilers currently in production, and a much larger flock of chickens fattened for market.

To find the population of the breeding flock at any given moment, we collected data of intended placements (the chicks who will become the next group of breeding hens) of broiler-type pullet chicks (young hens) from USDA National Agricultural Statistics Service (NASS) [41]. These placement numbers are rough estimates because broiler hatchery data is based on surveys from individual contractors and independent producers that were combined with historic trends [41]. Residence time is how many years a hen spends laying eggs for new broilers. Hens have an average production cycle of 40 weeks or 0.767 years [42,43]. To find the population of hens at any given moment in the conventional system, we multiplied the mean residence time of hens by the yearly rate at which hens are placed, which was assumed steady state:

$$p_{\text{breeding flock (conventional)}} = \tau_{\text{breeding flock (conventional)}} \times f_{\text{breeding flock placement (conventional)}}$$
$$= 0.767 \text{ yr} \times 98.0 \text{ million hens yr}^{-1}. \tag{2.1}$$

We then modelled the relatively larger flock of fattening chickens. The population of chickens at any given moment only accounts for a fraction of the total number of birds killed for meat each year in the United States. Conventional Ross chickens are slaughtered around 47 days of age or 0.129 years [28]. After assuming a steady state, we averaged the number of chickens placed every year from the number of chickens placed in 2018 and the number of chickens slaughtered in 2018 from the USDA NASS [1]. From here, we calculated the steady-state population of chickens in the conventional system by multiplying the mean residence time by the yearly rate at which chickens are placed:

$$p_{\text{fattening (conventional)}} = \tau_{\text{fattening (conventional)}} \times f_{\text{fattening lacement (conventional)}}$$
$$= 0.129 \text{ yr} \times 9.25 \text{ billion chickens yr}^{-1}. \tag{2.2}$$

The mean dressed weight (carcass weight after organs and blood are removed) of chickens was calculated by using average live weight from 2021 USDA NASS [27] and dressing percentage from averaging the 2.8 kg and 3.0 kg male and female eviscerated carcass percentage from Aviagen [26]. To determine the average dressed weight for each chicken, the mean live weight of chickens of 2.93 kg per chicken [27] was multiplied by the average dressing percentage of 74% [44]. The best mean dressed weight estimate for a faster-growing chicken is 2.17 kg per chicken.

By multiplying the average dressed weight of conventional Ross chickens with the steady-state placement rate, we calculated the kilograms of chickens produced per year.

## 2.2. Land requirements for the conventional system

The National Chicken Council states that approximately 20 000 birds are kept in a 1486 m$^2$ grow-out house [45]. If Ross are housed according to the NCC's description, the current space occupied by conventional Ross broilers can be calculated according to the following stocking rate:

$$\frac{s_{\text{current (conventional)}}}{p_{\text{fattening (conventional)}}} = 0.0743 \text{ m}^2 \text{ chicken}^{-1}, \tag{2.3}$$

where $s_{\text{current (conventional)}}$ is the space requirement for raising chickens in the conventional system and $p_{\text{fattening (conventional)}}$ is the population of Ross currently in the fattening system in CAFOs.

Part of Global Animal Partnership's certification includes outdoor access, which is not the same as organically raised chicken. These chickens live in stationary housing with seasonal outdoor access [18]. To raise Ross breeds with outdoor access, GAP requires that the outdoor area must be greater than or equal to 75% of the indoor floor space of the house. To find how much space would be needed if Ross were raised in CAFOs and with outdoor access, we multiplied the current direct land use of fast-growing breeds in CAFOs by 75% and added this to the assumed direct space to raise broiler chickens in CAFOs.

Some farms also raise chickens on pasture, which are also not the same as organically raised chickens. According to Humane Farm Animal Care, every 1000 birds raised on pasture must be given a minimum

of 0.0101 km$^2$ of space [34]. To find out how much space would be needed if conventional Ross chickens are raised on these requirements of pasture, we used the following stocking rate:

$$\frac{s_{\text{pasture (conventional)}}}{p_{\text{fattening (conventional)}}} = 10.12 \ \text{m}^2 \ \text{chicken}^{-1}. \tag{2.4}$$

The land that is used to grow feed for chickens must also be considered when quantifying the nationwide transformation. The NCC reported the following measurements of feed used in 2019: more than 1.4 billion bushels (35.56 million metric tons or MMT) of maize per year, more than 580 million bushels (15.78 MMT) of soybeans per year, and 64 million tons (58.1 MMT) of the total amount of mixed feed used [46,47]. From here, we calculated the average composition of chicken feed in terms of percentages for maize (60.3%), soybeans (26.8%) and other feed (12.9%). To find the land-use requirements for maize and soybeans in km$^2$ per year in the USA. (*Li*) we used the following equations using data from FAO [48]:

$$Li_{\text{km}^2 \text{ maize yr}^{-1} \text{ (conventional)}} = 35.56 \ \text{MMT maize yr}^{-1} \times 1 \ \text{ac} \ 4.802 \ \text{tonnes}^{-1} \text{maize} \tag{2.5}$$

and

$$Li_{\text{km}^2 \text{ soybeans yr}^{-1} \text{ (conventional)}} = 15.78 \ \text{MMT soybeans yr}^{-1} \times 1 \ \text{ac} \ 1.403 \ \text{tonnes}^{-1} \text{soybeans} \tag{2.6}$$

## 2.3. Demographics, live weight and cumulative feed for slower-growing breeds

We found that Aviagen tends to overestimate its Ross 308 broiler performance (feed requirements and slaughter weights) relative to the average performance reflected in USDA agricultural statistics. This overestimate was perhaps to due optimistic assumptions or from company animal scientists growing broilers in ideal conditions.

Data from Aviagen were used to calculate live weight of the slower-growing breeds, including the intermediate-growing Ranger Classic (RC) breed [49], the intermediate-growing Ranger Gold (RG) breed [50] and the slow-growing Rowan Ranger (RR) breed [51], but we scaled these live weights using average slaughter weights from the USDA, reflecting the conventional fast-growing Ross breed [26]. We divided USDA slaughter weight at 47 days by the Aviagen slaughter weight at 47 days to get a weight scale factor. We multiplied these scaled proportion for the RC, RG and RR breeds to find the appropriately scaled live slaughter weights for the three slower-growing breeds.

Because we wanted to find how many slower-growing chickens are needed to produce the same number of kilograms of chicken meat the USA currently produces, we worked backwards from the USDA total live chicken weight. We divided the USDA total live chicken weight by the three different scaled weight and dressing percentage (from Aviagen [49–51]) for each breed to determine the total number of chickens necessary to produce the same amount of chicken weight,

$$f_{\text{(breed)}} = \frac{W_{\text{(conventional)}}}{w_{\text{(breed scaled weight)}} \times 0.76}, \tag{2.7}$$

where $f_{\text{(breed)}}$ is the number of chickens slaughtered per year of each breed, $W_{\text{(conventional)}}$ is the current total live chicken weight and $w_{\text{(breed scaled weight)}}$ is the scaled live weight of each breed.

Because the population of chickens increases, the breeding flock must also increase. We assumed the new breeding flock population increases proportionally to the increased population of the three slower-growing breeds:

$$\frac{p_{\text{breeding flock (conventional)}}}{p_{\text{fattening (conventional)}}} = \frac{p_{\text{breeding flock (breed)}}}{p_{\text{fattening (breed)}}}, \tag{2.8}$$

where $p_{\text{breeding flock (conventional)}}$ is the population of hens in the breeding flock in CAFOs, $p_{\text{fattening (conventional)}}$ is the population of conventional Ross in the fattening stage, $p_{\text{breeding flock (breed)}}$ is the population of hens of each in the breeding flock, and $p_{\text{fattening (breed)}}$ is the population of each breed in the fattening stage.

Using the new population of the breeding flock, we adjusted for the turnover rate of hens needed according to the following equation:

$$f_{\text{breeding flock placement (breed)}} = \frac{p_{\text{breeding flock (breed)}}}{\tau_{\text{breeding flock (breed)}}}. \tag{2.9}$$

These calculations reflect the increase of population and residence time of each breed and the necessary population of the breeding flock to keep the production of chicken meat the same amount as current systems.

## 2.4. Land requirements for slower-growing breeds

If slower-growing breeds are to be kept in conventional grow-out houses, additional space will be required to account for the additional population of chickens, since the population increase is proportional to the spatial increase. We multiplied the CAFO space by the per cent increase of the new population to calculate the additional space needed to raise slower-growing chickens in grow-out houses.

To find out how much total space will be needed for a full population of chickens of each breed raised in CAFOs and with outdoor access, we multiplied the current direct space necessary to raise each breed in CAFOs by 75% and added this to the current direct land use of each of the breeds raised in CAFOs.

To find out how much direct space will be needed for a full population of each of the breeds raised on HFAC requirements of pasture, we used the equation:

$$\frac{S_{\text{pasture (breed)}}}{p_{\text{fattening (breed)}}} = 10.12 \ \text{m}^2 \ \text{chicken}^{-1}. \tag{2.10}$$

These spatial calculations only consider direct land use and do not include the increased land needed for increased feed.

For indirect land-use requirements for feed, slower-growing chickens have a different yearly feed quantity than Ross breeds. Data from Aviagen were used to calculate the cumulative feed intake of the intermediate-growing RC breed [49], the intermediate-growing RG breed [50] and the slow-growing RR breed [51] scaled from the conventional fast-growing Ross breed [26].

To adjust for possible overestimates in Aviagen slower-growing breed performance estimates, we calculated a 'performance scale factor'. We divided the cumulative feed per bird derived from the USDA agricultural statistics and nationwide NCC feed numbers. We then divided this by the Aviagen Ross cumulative feed per bird to find a performance scale factor of approximately 1.35, which accounts for the degree to which Aviagen appears to underestimate feed requirements when comparing with total US broiler performance and feed usage statistics in a top-down manner. We multiplied this performance scale factor by the Aviagen-reported feed requirements of each slower-growing breed. We then used the same average composition of chicken feed in terms of percentages for maize (60.3%), soybeans (26.8%) and other feed (12.9%) as Ross to find the average cropland requirements of maize and soybeans in km$^2$ per year in the USA. We conservatively assume that all 'other feed' is not crop-based, and therefore has no land-use requirements.

## 2.5. Changing consumption of slower-growing chicken breeds to not exceed current land use

For land use to stay the same, a nationwide shift to producing and consuming slower-growing breeds, chickens with outdoor access and/or pasture-raised chicken would require changes in consumption. To calculate this, we divided the land requirements for conventional Ross in CAFOs and divided this by the land requirements for each of the alternative breed and management scenarios (slower-growing breeds in CAFOs, each breed with outdoor access, and each breed grown on pastures). We then multiplied this with the $W_{\text{(conventional)}}$ to find the new weight of chicken meat produced per year for each of the scenarios:

$$W_{\text{(scenario)}} = \frac{\text{total land use}_{\text{(Ross in CAFOs)}}}{\text{total land use}_{\text{(scenario)}}} \times W_{\text{(conventional)}}. \tag{2.11}$$

How much consumption will decrease for each scenario was calculated using the following equations:

$$\text{decrease in kilograms} = W_{\text{(conventional)}} - W_{\text{(scenario)}} \tag{2.12}$$

and

$$\text{decrease in percentage} = 1 - \frac{\text{total land use}_{\text{(Ross in CAFOs)}}}{\text{total land use}_{\text{(scenario)}}} \times 100. \tag{2.13}$$

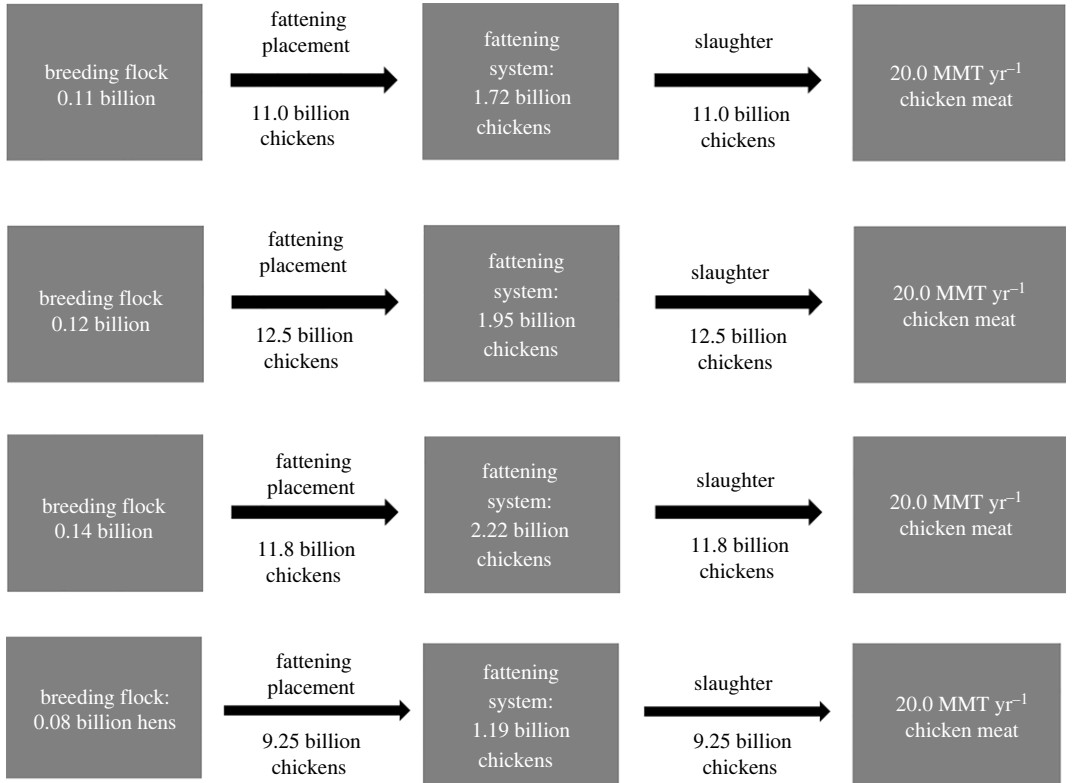

**Figure 1.** Rates of production and populations of broiler chickens in the conventional system in the USA in 2018, (A) Intermediate (RC), (B) Intermediate (RG), (C) Slow (RR).

To find the slaughter rate of chickens, we divided the kilograms of chicken consumed each year by the average dressed weight of each breed. Then we found the population by multiplying the slaughter rate by the average residence time of each breed.

## 3. Results

Each of the hypothetical parameters in the Methods section gets three different values for each respective breed: intermediate-growing RC and RG, and slow-growing RR.

### 3.1. Conventional chicken demographics and production

A simple box model of the current national demographics of broiler chickens is given in figure 1. The average annual population is 75.1 million breeding hens laying new broiler chicks (equation (2.1)) and 1.19 billion chickens in CAFOs (equation (2.2), figure 1). The current chicken CAFO system produces approximately 20 million metric tons of chicken meat each year.

Current US broiler chicken CAFO facilities occupy 88 km$^2$ of land directly (equation (2.3)). Indirect land use to grow feed for chickens is 75 488 km$^2$ of maize and soybean cropland (equations (2.5) and (2.6)). This number does not include the additional 6.7 MMT of other feed fed to chickens annually, much of which consists of mined minerals and fish by-products. The total space needed to raise and grow feed for conventional Ross on CAFOs, with outdoor access, or pasture is provided in figure 2 (equations (2.3), (2.4), (2.5) and (2.6)).

### 3.2. Demographics of slower growing breeds

The increase in chicken population, placements and slaughter rates for RC, RG and RR chicken breeds are shown in figure 1A, B and C. RC, RG and RR chickens have lower live slaughter weights by 0.409, 0.661 and 0.499 kg, respectively, and longer residence time by 10, 10 and 12 days respectively. The total populations will increase by 44.6%, 63.7% and 86.8%, respectively, to produce the same amount of

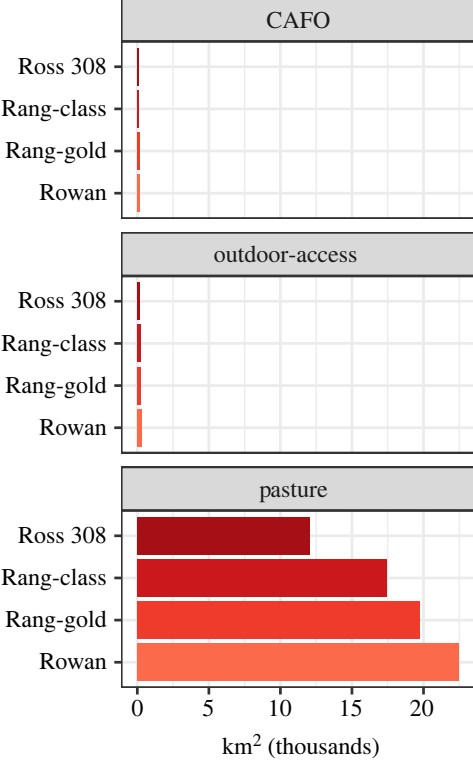

**Figure 2.** Direct land use in thousand km$^2$ required for varying breeds and management assuming consumption remains the same as present day.

chicken meat as the current fattening system. To reach the same amount of chicken meat as the current system, the RC, RG and RR annual placement and slaughter rate will increase by 19.22%, 35.0% and 27.2%, respectively.

The increased placement and slaughter rates of Ranger Classic and Ranger Gold breeds will also require an increase in each breed's breeding flock (equations (2.8) and (2.9); figure 1A and B). With this increased population of breeding hens, the population of intended placement of pullet chicks will increase from the current 98 million per year to 142 million, 160 million and 183 million RC, RG and RR hens (equation (2.9)).

## 3.3. Land requirements for slower-growing breeds

Currently, Ross 308 birds in CAFOs require 88 km$^2$ of total direct space for housing. The increased land use for a nationwide switch to RC, RG and RR breeds raised in CAFOs, with outdoor access, or on pasture is summarized in figure 2. To raise 20 million metric tons of RC, RG, and RR chicken meat on pasture each year, there would need to be 17 412 km$^2$, 19 719 km$^2$ and 22 498 km$^2$ direct space to house the larger populations of slower-growing breeds at any given time (equation (2.10)). These increases in land use are approximate since these minimum spatial requirements are recommendations from one organization, HFAC. To raise 20 million metric tons of the RC, RG and RR breeds with outdoor access each year, there will need to be a total of approximately 224 km$^2$, 254 km$^2$ and 289 km$^2$ direct space, respectively.

Because slower-growing breeds also have lower feed conversion ratios (FCR), indirect land use to meet nutrient requirements will also increase. Indirect land for slower-growing chickens is outlined in figure 3 (equations (2.5) and (2.6)). This does not include the 13% of 'other' feed requirements provided in USDA estimates, which could increase indirect land use feeds further provided more precise data on these feed sources becomes available. Figure 3, therefore, includes both the total direct and indirect space needed to raise and grow feed for slower-growing breeds (equations (2.5), and (2.6)) because we conservatively did not assume that changing the housing management to outdoor access or pasture would entail different nutrient requirements. Conventional Ross 308 breeds require a total of 75 488 km$^2$ of indirect land for

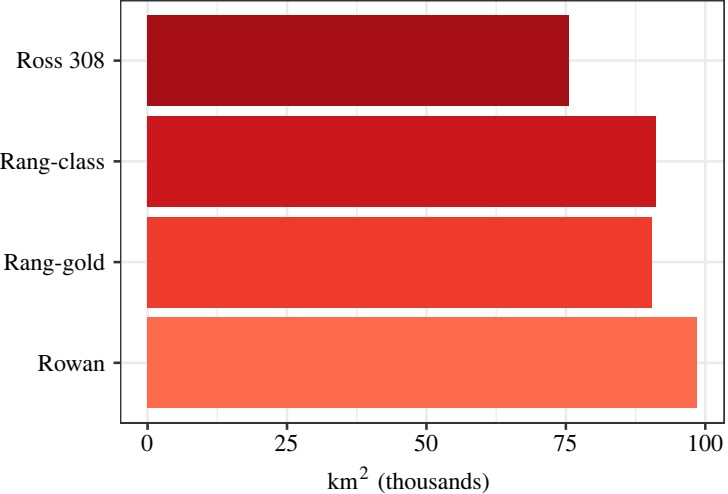

**Figure 3.** Indirect land use for crop production in thousand km$^2$ required for varying breeds assuming consumption remains the same as present day. Per assumptions in this analysis, management style does not change feed requirements and resulting indirect land use.

predominantly corn and soy feed. Switching to alternative breeds would require 91 229 km$^2$, 90 437 km$^2$ and 98 521 km$^2$ for breeds RC, RG and RR, respectively.

## 3.4. Changes in consumption necessary to not exceed current land use

For the total space to raise slow-growing, intermediate breeds, pasture-raised chickens and/or give outdoor access to stay the same as current conventional systems, total direct plus indirect land use cannot exceed 75 577 km$^2$. This necessitates decreases in total nationwide consumption by 3.46 million metric tons (17.3%) less for intermediate RC breeds raised in CAFOs, 3.32 million metric tons (16.7%) less for intermediate RG breeds raised in CAFOs, 4.69 million metric tons (23.4%) less for slow-growing RR breeds raised in CAFOs, 2.74 million metric tons (13.7%) less for Ross raised on pasture, 6.1 million metric tons (30.4%) less for intermediate RC breeds raised on pasture, 6.29 million metric tons (31.4%) less for intermediate RG breeds on pasture, 7.52 million metric tons (37.6%) less for slow-growing RR breeds on pasture, 17.6 million kg (0.0%) less for Ross with outdoor access, 3.48 million metric tons less (17.4%) for intermediate RC breeds with outdoor access, 3.34 million metric tons (16.7%) less for intermediate RG with outdoor access, and 4.71 million metric tons (23.5%) less for slow-growing RR breeds with outdoor access (table 1). The maximum chicken consumption, annual slaughter rates of chickens and population of chickens if present-day total land use is held constant is outlined in table 1 for all scenarios.

Compared with the current conventional system of Ross 308 chickens farmed in CAFOs, our models holding land use constant suggest that providing chickens with some outdoor access would necessitate very little change to consumption patterns or slaughter rates (figure 4). All other production scenarios would require substantial reductions in consumption, with Rowan breeds raised on pasture requiring the greatest reduction in consumption from the current value of 20 MMT with Ross 308 in CAFOs to 12.5 MMT with Rowan on pasture, a 37% cut in overall US chicken consumption. The Rowan-pasture scenario also had the lowest annual slaughter rates of 7.35 billion chickens yr$^{-1}$, but importantly, this scenario would also involve a 27% increase in the US chicken population at any given moment, from 1.19 billion Ross 308 chickens currently in CAFOs to 2.22 billion Rowan chickens living on pasture (figure 5).

# 4. Discussion and conclusion

Our study represents an interdisciplinary One Health approach that simultaneously models changes to demographics—a prerequisite for aggregate welfare considerations, along with land-use requirements.

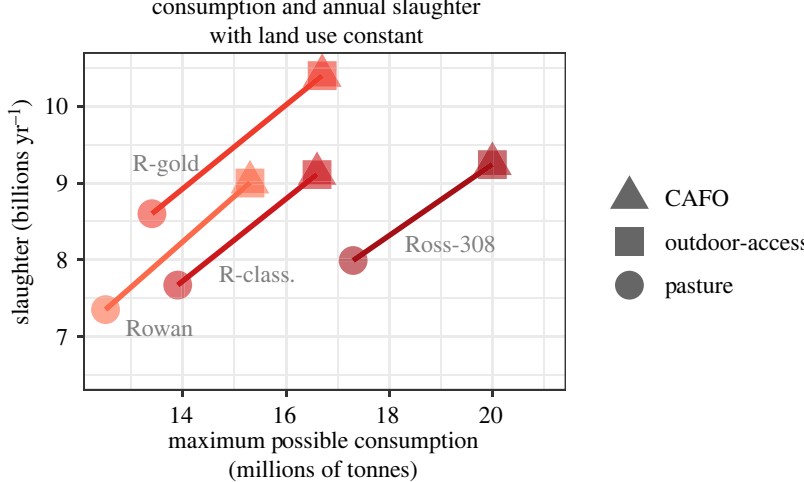

**Figure 4.** Consumption and annual slaughter rate of broiler chickens required to hold total land use constant (direct + indirect).

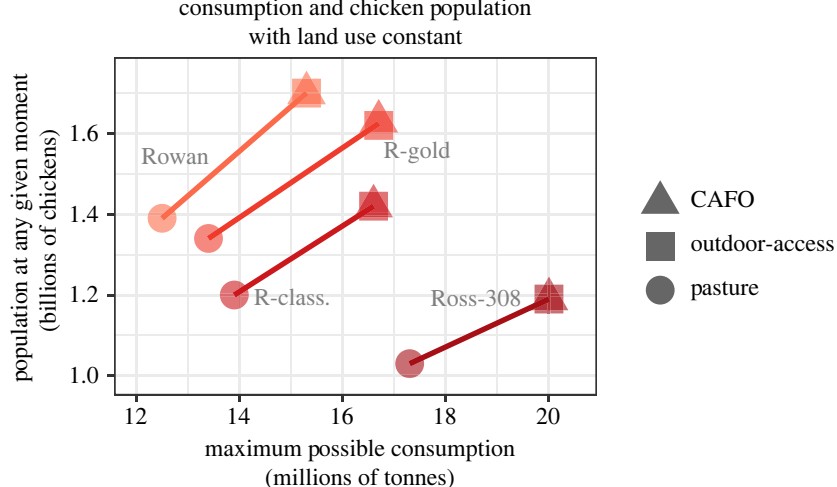

**Figure 5.** Consumption and average population of broiler chickens required to hold total land use constant (direct + indirect).

**Table 1.** Maximum chicken consumption, annual slaughter rates of chickens and population of chickens if present-day total land use is held constant (75 577 km$^2$).

|  | maximum chicken consumption (million metric tons) | annual slaughter rates of chickens (billion chickens) | population of chickens (billion chickens) |
| --- | --- | --- | --- |
| Ross in CAFOs | 20.0 | 9.25 | 1.19 |
| Ross with outdoor access | 20.0 | 9.24 | 1.19 |
| Ross on pasture | 17.3 | 7.99 | 1.03 |
| RC in CAFOs | 16.6 | 9.12 | 1.42 |
| RC with outdoor access | 16.6 | 9.11 | 1.42 |
| RC on pasture | 13.9 | 7.67 | 1.20 |
| RG in CAFOs | 16.7 | 10.4 | 1.63 |
| RG with outdoor access | 16.7 | 10.4 | 1.62 |
| RG on pasture | 13.7 | 8.60 | 1.34 |
| RR in CAFOs | 15.3 | 9.01 | 1.70 |
| RR with outdoor access | 15.3 | 9.00 | 1.70 |
| RR on pasture | 12.5 | 7.35 | 1.39 |

When these elements are considered in tandem, important and complicated trade-offs emerge that are not present when considering efficiency, welfare or land use in isolation. Through analysing a range of breeds and management systems, we provide a range of estimates for feed requirements, population and hence land use for shifts intended to improve broiler chicken welfare. For land use, notably, we provide a conservative estimate for indoor housing and a high-space form of commercial-scale pasture management. Shifts in management that hybridize these two methods, such as partial outdoor access that transitions to pasture in the final weeks of maturity, would land between these two outcomes. By examining a range of outcomes, we have attempted to quantify the 'sustainability gap' entailed by shifting to alternative breeds and management that are considered by many to be more humane and sustainable. Conversely, we quantify the operating space or thresholds that chicken meat production and consumption should not exceed if land use or populations of chickens are held constant.

Although raising slower-growing breeds such as Rangers may improve individual animal welfare, the raising, managing, transporting and slaughtering of additional chickens required by shifts to alternative production could impact aggregate chicken welfare at the country-level scale, that is, how many chickens experience stressful conditions. Our results indicate that, if raised in CAFOs, a shift to slower-growing Rangers may increase crowding and related welfare concerns including increased footpad dermatitis [7], jostling, conflicts and potentially infection risk [14], and thus may translate to a decrease in aggregate welfare at scale. A shift to individual better-welfare chicken breeds aims to lessen bone, heart and disease issues in present Ross birds, but even in non-CAFO production systems, slower-growing breeds may still experience other negative welfare conditions such as emotional and physical stress, disease, predation, injury and premature mortality, as well as distressing transport and slaughter practices [52–54]. While some of these negative consequences could be mitigated by an increase in positive experiences (e.g. increased opportunities for exploration and choice and more positive emotions), without dramatic improvements to housing and management systems and attendant legislation to ensure their implementation, the added value of positive experiences is unlikely to outweigh the cumulative negative impacts of large-scale production. In the USA in particular, the Humane Slaughter Act and Humane Welfare act do not provide even basic protections for poultry because they are not legally categorized as 'livestock'. These conditions and a systemic lack of their regulation potentially justify an assumption that chickens in commercial production, even if breeds and management are improved, exist in a largely negative-welfare state, but more analysis and discussion of aggregate welfare are needed beyond this analysis. At a minimum, because shifts to slower-growing breeds entail creating greater populations and increasing annual slaughter rates, such shifts have uncertain consequences for aggregated welfare at scale.

Shifts to pasture-based systems also entail welfare trade-offs. Pasture-based systems reduce stocking rates, thus reducing crowding. An increase in space leads to more active chickens, reducing excessive muscle strains [35,36]. However, there remain significant welfare implications for raising slower-growing chickens on more spacious outdoor management systems. Although birds raised on pasture can move freely all or part of the day whether in a fenced-off area or the open field, chickens face the risk of natural predation by hawks or snakes [38]. Furthermore, additional land for raising chickens on pasture as well as croplands will potentially damage surrounding environments, disrupt local forest habitats and wildlife, and displace ecosystem carbon [39] and biodiversity [40]. Biodiversity loss can negatively affect human wellbeing, leading to further unintended consequences [25].

Our analysis does not take into account the consequences of differences between the breeds or management systems in terms of mortality during production or carcass rejection rates at slaughter. Mortality and carcass rejection rate could be reasoned to go up or down in alternative production systems. On the one hand, the better welfare breeds on pasture tend to have lower stress and aggression, hence improved immune systems, potentially leading to decreased mortality and carcass rejection. On the other hand, alternative breeds on pasture may face physical injuries, parasites, insect bites or predation, potentially leading to increased mortality and carcass rejection. In the absence of research indicating the likely net influence on mortality and carcass rejection, we omitted these considerations from this analysis—crucial parameters that could ultimately impact the larger picture and thus require further study.

To support the annual quantity of chicken meat produced in the USA, pasture-raised system requirements generally demand more land than current CAFOs. However, there exists a number of relatively minor but nonetheless consequential trade-offs with respect to feed requirements, hence land use, between CAFOs and pastures, which are challenging to quantify and for which poor or no nationally representative data exists. Slower-growing breeds may have lower protein requirements than the conventional Ross breed. Also, while pasture-based system FCRs and live weight are not as

optimal as those in conventional systems, due to increased activity hence higher metabolic demands [35], pasture-raised birds supplement their grain-based diets through pecking and foraging. Insect and other supplemental pasture-based feed quality may in turn vary widely between locations of potential pasture. Additionally, we assume that pasture is entirely 'open' and did not test assumptions regarding the level of vegetation cover and mix on pastures; bushes and trees can provide shelter for chickens and grubs along with ecosystem services but may also increase open-space requirements if the area is too crowded with such natural vegetation. There also exists a production trade-off of a decreased mortality rate for slower-growing birds [7,29]. These issues all warrant further analysis to more precisely quantify feed efficiency and productivity trade-offs as our knowledge of feed and land-use responses to changing breed and management variables improves.

This study demonstrates that the annual slaughter rate of chickens must decrease for a successful nationwide shift to a slower-growing breed of chickens on pasture (table 1). The population of chickens at any one given moment will vary depending on the chicken breed and the living conditions (table 1). To achieve a nationwide shift of intermediate RC chickens on pasture without increasing land use, for example, there must be a decrease in consumption of chickens from the current 20 million metric tons by 6.1 million metric tons (table 1), a reduction of 30.4%. Smaller changes in consumption are necessary if either only the breed of chicken or only the management system is changed.

The concept of One Health interconnects animal welfare, the environment and human behaviour [24,25]. While key outcomes have previously been analysed for changing housing systems or breeds, food production impacts should be gauged not just in one stage of production throughout the entire system of production, processing, marketing and consumption [55]. While seeking improved welfare for individual chickens, shifts to slower-growing breeds entail significant increases in chicken populations and slaughter rates required. Such shifts could have uncertain consequences for animal welfare overall when aggregated welfare is considered. Furthermore, shifts to slower-growing breeds and/or pasture management entail expansion and intrusion into unfarmed native ecosystems, which would negatively impact wild animal welfare and species biodiversity if consumption does not decrease in tandem. Additionally, chicken processing requires humans to kill the chickens, causing physical illnesses and injuries and psychological damage to the workers [56]. Human decisions regarding how much chicken they consume, which breed of chicken is raised, how the chickens are housed and reared all influence the number of chickens slaughtered. The One Health concept stresses that human, animal and environmental health and welfare are all interdependent, further illuminated by cascading changes in welfare, land use and consumption necessitated by alternative chicken production.

Our analysis scales its focus from individual animals to other notable metrics for understanding animal welfare and the environmental impacts at the country-level scale. It is our intention that these results highlight opportunities for scientists working in areas of nutrition and performance, environment and animal welfare to work in tandem along with civil society, to tackle important trade-offs between welfare and land usage and its attendant impacts.

Our results detail for the first time constraints and trade-offs relevant to chicken welfare, environmental conservation and human decision-making. We thus aim to provide valuable policy-relevant information in addition to contributing to our understanding of One Health and informing future work in related systems. Greatly decreasing chicken consumption could reduce slaughter rates, population sizes and land requirements entailed by shifts to slower-growing breeds of chickens and more spacious management systems, avoiding further displacement of wild habitats and improving aggregate welfare. Reduced chicken consumption thus avoids potential trade-offs between welfare and environmental protection that this analysis has highlighted. Several studies to date have demonstrated effective strategies of limiting meat consumption, including: reducing portion sizes [57]; shifting some meals toward nutritionally complete vegetarian meals [57]; incorporating social media and online platforms to strategize meat reduction campaigns that provide education, information and knowledge on recipes and nutrition [58]; encouraging political action [59]; and partnering with the food industry to provide more plant-based choices [59].

Data accessibility. The data used for the calculations and figures in this manuscript can be found in the supplementary data, at the Dryad Digital Repository: https://doi.org/10.5061/dryad.d2547d82g [60].

Authors' contributions. I.C.: conceptualization, data curation, formal analysis, funding acquisition, investigation, methodology, project administration, writing—original draft, writing—review and editing; B.F.: conceptualization, data curation, formal analysis, writing—original draft, writing—review and editing; M.N.H.: conceptualization, data curation, formal analysis, methodology, writing—original draft, writing—review and editing.

All authors gave final approval for publication and agreed to be held accountable for the work performed therein.

Conflict of interest declaration. The authors have declared that no competing interests exist.

Funding. I.C. received an award from the Centre for Environmental and Animal Protection (CEAP: https://wp.nyu.edu/ceap/) at New York University. The funder had no role in study design, data collection and analysis, decision to publish or preparation of the manuscript.

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
