## [Peer Review File · Royal Society Open Science]

Review History

RSOS-210478.R0 (Original submission)

Review form: Reviewer 1

Is the manuscript scientifically sound in its present form?

No

Are the interpretations and conclusions justified by the results?

No

Is the language acceptable?

Yes

Do you have any ethical concerns with this paper?

No

Have you any concerns about statistical analyses in this paper?

No

Recommendation?

Major revision is needed (please make suggestions in comments)

Comments to the Author(s)

Attached (see Appendix A).

Review form: Reviewer 2**Is the manuscript scientifically sound in its present form?**

Yes

Are the interpretations and conclusions justified by the results?

Yes

Is the language acceptable?

Yes

Do you have any ethical concerns with this paper?

Yes

Have you any concerns about statistical analyses in this paper?

No

Recommendation?

Major revision is needed (please make suggestions in comments)

Comments to the Author(s)

It is the overall impression that the manuscript contains interesting information since welfare and environmental issues are important in any animal production.

Since it is an American study, the legislation regarding production of conventional broiler chickens is probably different from the European, so I find it important that there are some information about the American legislation given in the text. Further, I find part of the Introduction very subjective in the description of the conventional broiler production, which probably gives a skewed view of the production compared to alternative productions. More concrete information should be given to explain in an objective way the conventional and alternative broiler productions in the United States.

Introduction:

It is important to make it clear in the text that the paper is dealing with broiler chickens for meat production and not chickens (pullets) for egg production and if it is the conventional production or an alternative production. Therefore, it should be written already in the first line Page 3, 38: In 2018, the conventional broiler chicken production (delete industry) ----- In line 43 delete known as broilers.

Line 40: suggests: as the demand for chicken meat production increases, so too---: what does this mean? It is not very clear: who have concerns - the consumers?

Line 42: after environment, such as (list some few examples).

Line 44-45: by mentioning these practices, the conventional broiler production appear very negative and is it really correct that broiler chickens are beak trimmed? Due to their very short life span, this is not normal practise (at least in Europe) and therefore an unnecessary cost for the farmer. Feather pecking is mainly seen with some layer genotypes having a long production cycles. The reference Nicol, C (2018) is dealing with egg layers and not broilers and it also appear

to be the case for the reference by Schwean-Larden, K. It is suggested that the authors find more relevant references, and inform about the practice for broilers and not layers, since these productions are very different. This is actually mentioned by the authors on page 7, line 120-121. Stocking densities: what are the legislation for stocking densities? For instance in kg/pounds per m² at slaughter age. If there is no legislation in this area, it should also be mentioned.

Line 46: cannibalism – the references are related to pullets/layers and not broilers. Cannibalism is not seen very often in broiler production, and at least references should deal with cannibalism in broilers.

Page 4: in published papers, presentation of a topic should be objective and academic, and therefore words, such as cruelty, should be avoided, or if it is a statement from e.g. an animal welfare organisation, it could be given as a citation “----” and with a reference included.

Line 49-50. Suggest to change the text to: --- alternative ways to produce broiler chickens from the conventional, intensive system ----.

Line 60-62: is it really common practice to have a slaughter age at 47 days??? Ross 308 is very fast growing and continuing the production until 47 days will be very expensive in feed costs and the chickens will probably not be able to walk very well the last 1-2 weeks, which increase the risk for wounds and breast blisters, since the chickens will sit down on the litter more often. This could result in a higher discarding percentage at the slaughter house, which reduce the payment to the farmer. Is there any information about these data available? If some kind of feed restriction is used to reduce weight gain, it should be mentioned. In the recent Ross 308 management guide (2021) from Aviagen the final live weight at 47 days is 3,75kg (8,33pounds), which are in line with the data given on market weight, so probably, the chickens have been feed ad. lib.

Line 66: higher instead of greater

Line 69-71. Some of the parameters mentions are repetitions from line 67-69.

Page 6, line 95: regarding health consequences, references should be found dealing with e.g. risk of coccidiosis or at least mention the importance of different vaccinations. The ref. Chen et al (2013) is more dealing with the effect of outdoor area on meat quality and not about potential risk for pathogen infections. This information should be added to the text.

Line 99: write nutrient requirement instead of feed requirement.

Line 105: add conventional broiler before chickens

Methods

This paragraph includes a lot of equations for calculating the different results needed to discuss the topic and the explanation given is overall sufficient and understandable.

Line 164-165: what is the difference between the current conventional system and CAFO's? CAFO's are defined on page 3, line 43, but it is not clear how the current conventional system differs?

Line 168: it is probably 0.0101 km². Why not give the data also in m² per chicken?

Line 172 ----: a general comment: it should be mentioned somewhere that the alternative, outdoor systems are free-range, but not organic?? Production of organic feed would give another dimension to the calculations, which is probably not the case in this study, but it should be mentioned that the alternative systems considered are based on conventional feed ingredients. The use of the outdoor area by the broilers is very much dependant on the vegetation (trees and bushes/ or not) and the genotypes. Is this considered or is it “defined” that the broilers will use the total area regardless of the extent of planting?

Results: the description and explanation of the results appear sufficient, however, with a lot of information placed in three figures and one table. It is helpful that the equations used for the different calculations are included in brackets in each paragraph.

Discussion.

Page 16, line 314-315: How to define animal welfarists?? There are many animal scientist that have focussed on how to improve welfare for animals in different systems, Scientists working

with nutrition also look into alternative feeding strategies and the use of new protein sources for a more sustainable production, so this “statement”/sentence should be reformulated. So, scientists working with nutrition and performance and scientist working with welfare issues. Like the Introduction, it is important to be objective.

Line: it is very good to introduce the One Health/Welfare approach, but the cooperation between nutrition/performance and welfare disciplines are not new.

Page 17: line 339-341: Nonetheless, ----, for what reasons? It is not clearly explained? If the broilers have sufficient outdoor space, where they can express their natural behaviour, what factors will results in more stress and mortality: predation? This situation will probably be dependant of the quality of the outdoor area with regard to the amount of vegetation and areas with trees, where the broilers can seek protection against raptors. This is described more in details in the following lines 342-352, which is relevant information, but the lines 339-341, still appear confusing.

The discussion from line 361 --- regarding the needed change in how humans consume meat is very important on order to deal with the environmental and welfare issues for the animals, bur also regarding health and welfare for humans.

Decision letter (RSOS-210478.R0)

Dear Professor Chan,

The Editors assigned to your paper RSOS-210478 "Environmental and welfare consequences of alternative US chicken production" have now received comments from reviewers and would like you to revise the paper in accordance with the reviewer comments and any comments from the Editors. Please note this decision does not guarantee eventual acceptance.

Please submit your revised manuscript and required files (see below) no later than 21 days from today's (ie 27-Aug-2021) date. Note: the ScholarOne system will 'lock' if submission of the revision is attempted 21 or more days after the deadline. If you do not think you will be able to meet this deadline please contact the editorial office immediately.

on behalf of Professor Pete Smith (Subject Editor)
openscience@royalsociety.org

Associate Editor Comments to Author:

Extensive feedback has been provided by two referees that you need to address before the paper may be considered acceptable for publication. Please carefully respond to and incorporate the changes recommended by the reviewers - you should provide a point-by-point response for each item identified by the reviewers (especially if you are rebutting any of their suggestions) and also a tracked changes version of the manuscript when you submit the revision. Good luck!

Reviewer comments to Author:

Reviewer: 1

Comments to the Author(s)

Attached (see file "RSOS broiler review.pdf").

Reviewer: 2

Comments to the Author(s)

It is the overall impression that the manuscript contains interesting information since welfare and environmental issues are important in any animal production.

Since it is an American study, the legislation regarding production of conventional broiler chickens is probably different from the European, so I find it important that there are some information about the American legislation given in the text. Further, I find part of the Introduction very subjective in the description of the conventional broiler production, which probably gives a skewed view of the production compared to alternative productions. More concrete information should be given to explain in an objective way the conventional and alternative broiler productions in the United States.

Introduction:

It is important to make it clear in the text that the paper is dealing with broiler chickens for meat production and not chickens (pullets) for egg production and if it is the conventional production or an alternative production. Therefore, it should be written already in the first line Page 3, 38: In 2018, the conventional broiler chicken production (delete industry) ----. In line 43 delete known as broilers.

Line 40: suggests: as the demand for chicken meat production increases, so too---: what does this mean? It is not very clear: who have concerns - the consumers?

Line 42: after environment, such as (list some few examples).

Line 44-45: by mentioning these practices, the conventional broiler production appear very negative and is it really correct that broiler chickens are beak trimmed? Due to their very short life span, this is not normal practise (at least in Europe) and therefore an unnecessary cost for the farmer. Feather pecking is mainly seen with some layer genotypes having a long production cycles. The reference Nicol, C (2018) is dealing with egg layers and not broilers and it also appear to be the case for the reference by Schwean-Larden, K. It is suggested that the authors find more relevant references, and inform about the practice for broilers and not layers, since these productions are very different. This is actually mentioned by the authors on page 7, line 120-121.

Stocking densities: what are the legislation for stocking densities? For instance in kg/pounds per m² at slaughter age. If there is no legislation in this area, it should also be mentioned.

Line 46: cannibalism – the references are related to pullets/layers and not broilers. Cannibalism is not seen very often in broiler production, and at least references should deal with cannibalism in broilers.

Page 4: in published papers, presentation of a topic should be objective and academic, and therefore words, such as cruelty, should be avoided, or if it is a statement from e.g. an animal welfare organisation, it could be given as a citation “----” and with a reference included.

Line 49-50. Suggest to change the text to: --- alternative ways to produce broiler chickens from the conventional, intensive system ----.

Line 60-62: is it really common practice to have a slaughter age at 47 days??? Ross 308 is very fast growing and continuing the production until 47 days will be very expensive in feed costs and the chickens will probably not be able to walk very well the last 1-2 weeks, which increase the risk for wounds and breast blisters, since the chickens will sit down on the litter more often. This could result in a higher discarding percentage at the slaughter house, which reduce the payment to the farmer. Is there any information about these data available? If some kind of feed restriction is used to reduce weight gain, it should be mentioned. In the recent Ross 308 management guide (2021) from Aviagen the final live weight at 47 days is 3,75kg (8,33pounds), which are in line with the data given on market weight, so probably, the chickens have been feed ad. lib.

Line 66: higher instead of greater

Line 69-71. Some of the parameters mentions are repetitions from line 67-69.

Page 6, line 95: regarding health consequences, references should be found dealing with e.g. risk of coccidiosis or at least mention the importance of different vaccinations. The ref. Chen et al (2013) is more dealing with the effect of outdoor area on meat quality and not about potential risk for pathogen infections. This information should be added to the text.

Line 99: write nutrient requirement instead of feed requirement.

Line 105: add conventional broiler before chickens

Methods

This paragraph includes a lot of equations for calculating the different results needed to discuss the topic and the explanation given is overall sufficient and understandable.

Line 164-165: what is the difference between the current conventional system and CAFO's? CAFO's are defined on page 3, line 43, but it is not clear how the current conventional system differs?

Line 168: it is probably 0.0101 km². Why not give the data also in m² per chicken?

Line 172 ----: a general comment: it should be mentioned somewhere that the alternative, outdoor systems are free-range, but not organic?? Production of organic feed would give another dimension to the calculations, which is probably not the case in this study, but it should be mentioned that the alternative systems considered are based on conventional feed ingredients. The use of the outdoor area by the broilers is very much dependant on the vegetation (trees and bushes/ or not) and the genotypes. Is this considered or is it “defined” that the broilers will use the total area regardless of the extent of planting?

Results: the description and explanation of the results appear sufficient, however, with a lot of information placed in three figures and one table. It is helpful that the equations used for the different calculations are included in brackets in each paragraph.

Discussion.

Page 16, line 314-315: How to define animal welfarists?? There are many animal scientist that have focussed on how to improve welfare for animals in different systems, Scientists working with nutrition also look into alternative feeding strategies and the use of new protein sources for a more sustainable production, so this “statement”/sentence should be reformulated. So,

scientists working with nutrition and performance and scientist working with welfare issues. Like the Introduction, it is important to be objective.

Line: it is very good to introduce the One Health/Welfare approach, but the cooperation between nutrition/performance and welfare disciplines are not new.

Page 17: line 339-341: Nonetheless, ----, for what reasons? It is not clearly explained? If the broilers have sufficient outdoor space, where they can express their natural behaviour, what factors will results in more stress and mortality: predation? This situation will probably be dependant of the quality of the outdoor area with regard to the amount of vegetation and areas with trees, where the broilers can seek protection against raptors. This is described more in details in the following lines 342-352, which is relevant information, but the lines 339-341, still appear confusing.

The discussion from line 361 --- regarding the needed change in how humans consume meat is very important on order to deal with the environmental and welfare issues for the animals, bur also regarding health and welfare for humans.

===PREPARING YOUR MANUSCRIPT===

===PREPARING YOUR REVISION IN SCHOLARONE===

Author's Response to Decision Letter for (RSOS-210478.R0)

See Appendix B.

RSOS-210478.R1 (Revision)

Review form: Reviewer 2

Is the manuscript scientifically sound in its present form?

Yes

Are the interpretations and conclusions justified by the results?

Yes

Is the language acceptable?

Yes

Do you have any ethical concerns with this paper?

No

Have you any concerns about statistical analyses in this paper?

No

Recommendation?

Accept with minor revision (please list in comments)

Comments to the Author(s)

In this corrected version of the manuscript (Manuscript ID RSOS-210478.R1), you have followed most of the recommendations given for improvement of the paper and in the cases, where you haven't agreed on a suggestion for changes, there are given good arguments for not including the correction: example: Rewiever: The discussion from line 361 --- regarding the needed change in how humans consume meat is very important on order to deal with the environmental and welfare issues for the animals, but also regarding health and welfare for humans. You have given a justified explanation for NOT including this aspect. The corrected paper has been improved to a large extent and when corrections are accepted in a final version of the manuscript, a thorough evaluation of the text has to be carried out for spelling errors and for a more fluent language, before publication, if finally accepted.

Review form: Reviewer 3

Is the manuscript scientifically sound in its present form?

No

Are the interpretations and conclusions justified by the results?

No

Is the language acceptable?

Yes

Do you have any ethical concerns with this paper?

No

Have you any concerns about statistical analyses in this paper?

No

Recommendation?

Major revision is needed (please make suggestions in comments)

Comments to the Author(s)

This is a very interesting and thought-provoking study, with potentially far-reaching implications. These consequences on the annual slaughter rate differences between breeds are almost exclusively caused by differences in slaughter weight between fast and slower growing breeds. This needs emphasizing because in other parts of the world, I think, these differences in slaughter weight can be very different. I'm from Europe, and Ross 308 tend to be slaughtered at a much lower weight and I think slower growing breeds reach more or less the same slaughter weight as the Ross 308. I presume that the authors have thoroughly checked the slaughter weights they have used for their calculations. In any case they ought to limit their conclusions to parts of the world where differences in slaughter weight are similar and perhaps point out that this may be very different in other parts of the world. If slower-growing breeds would be slaughtered at the same weight, than aggregate welfare would not or hardly been affected according to this study. It surprises me that in these calculations the dressing percentage seems to have been kept constant between breeds (and housing conditions). Is there evidence to support that dressing percentage is indeed not affected by breed nor by outdoor access? Apart from differences in mortality (which aren't taken into account and "dealt with" a bit easily in the Discussion, L464-467), differences in carcass rejection rates (at slaughter) may also influence the outcomes, but these do not seem to have been taken into account or are assumed not to differ between breeds or housing conditions. Again, is there evidence for this assumption?

For aggregate animal welfare, the annual slaughter rate is of crucial importance but not the size of the broiler population at any moment in time. This should be explained better and terminology to differentiate both metrics should be clarified from the very beginning, and these terms should subsequently be used more consistently throughout the manuscript (now terms such population, full population, total population, placements etc... are used and confuse the reader).

For aggregate animal welfare the authors seem to have a one-sided focus on negative welfare outcomes (L99-102, L423-441). L429-432 emphasizes this view and really ought to be better substantiated and references are needed to support this statement. Indeed the longer an animal lives and the more animals that need to be slaughtered, the higher the number of possibly negative experiences (all other things being kept equal). However, also the number of positive experiences will increase. The authors seem to assume that such positive experiences are much less common as compared to the negative ones, but this ratio might perhaps become positive in case of slower-growing breeds with outdoor access. If so, wouldn't aggregate welfare increase (rather than decrease) the more birds are kept per year and the longer they live? Perhaps this could be elaborated upon.

For the environmental impact, it seems that the feed composition is assumed not to differ depending on breed or housing condition. I'm not a nutritionist, but I'm pretty sure that slower-growing breeds need less protein-rich diets as compared to fast growing breeds. This probably implies that less soy is needed, which I would think has an important impact on the ecological footprint and land-use.

L402-404 I consider animal welfare science to be part of animal science. I don't agree that in animal welfare science only the individual is taken into account. To the contrary, most animal welfare monitoring protocols (e.g. Welfare Quality, AWIN,...) aim to assess the welfare of the group rather than the individual.

Decision letter (RSOS-210478.R1)

Dear Professor Chan

The Editors assigned to your paper RSOS-210478.R1 "The "sustainability gap" of US broiler chicken production: trade-offs between welfare, land-use, and consumption" have now received comments from reviewers and would like you to revise the paper in accordance with the reviewer comments and any comments from the Editors. Please note this decision does not guarantee eventual acceptance.

Please submit your revised manuscript and required files (see below) no later than 21 days from today's (ie 28-Feb-2022) date. Note: the ScholarOne system will 'lock' if submission of the revision is attempted 21 or more days after the deadline. If you do not think you will be able to meet this deadline please contact the editorial office immediately.

on behalf of Professor Pete Smith (Subject Editor)
openscience@royalsociety.org

Associate Editor Comments to Author:

Thanks to the reviewers who have provided comments on this revision - unfortunately, only one of the original referees was available to assist, and so we've received comments for a third reviewer. Given the concerns of the third reviewer, we are offering you the opportunity to perform further revisions - however, please be aware that, firstly, if you are not able to persuade

this reviewer that the manuscript is ready for acceptance after revision, we will not be able to accept it. Secondly, we recognise that you needed additional time to revise the initial submission, and we will give you a 6-week revision window to complete the revisions - however, no additional extensions will be permitted. If you are not able to complete the revisions in this timeline, you will have to resubmit the paper afresh.

Reviewer comments to Author:

Reviewer: 2

Comments to the Author(s)

In this corrected version of the manuscript (Manuscript ID RSOS-210478.R1), you have followed most of the recommendations given for improvement of the paper and in the cases, where you haven't agreed on a suggestion for changes, there are given good arguments for not including the correction: example: Rewiever: The discussion from line 361 --- regarding the needed change in how humans consume meat is very important on order to deal with the environmental and welfare issues for the animals, but also regarding health and welfare for humans. You have given a justified explanation for NOT including this aspect. The corrected paper has been improved to a large extent and when corrections are accepted in a final version of the manuscript, a thorough evaluation of the text has to be carried out for spelling errors and for a more fluent language, before publication, if finally accepted.

Reviewer: 3

Comments to the Author(s)

This is a very interesting and thought-provoking study, with potentially far-reaching implications. These consequences on the annual slaughter rate differences between breeds are almost exclusively caused by differences in slaughter weight between fast and slower growing breeds. This needs emphasizing because in other parts of the world, I think, these differences in slaughter weight can be very different. I'm from Europe, and Ross 308 tend to be slaughtered at a much lower weight and I think slower growing breeds reach more or less the same slaughter weight as the Ross 308. I presume that the authors have thoroughly checked the slaughter weights they have used for their calculations. In any case they ought to limit their conclusions to parts of the world where differences in slaughter weight are similar and perhaps point out that this may be very different in other parts of the world. If slower-growing breeds would be slaughtered at the same weight, than aggregate welfare would not or hardly been affected according to this study. It surprises me that in these calculations the dressing percentage seems to have been kept constant between breeds (and housing conditions). Is there evidence to support that dressing percentage is indeed not affected by breed nor by outdoor access? Apart from differences in mortality (which aren't taken into account and "dealt with" a bit easily in the Discussion, L464-467), differences in carcass rejection rates (at slaughter) may also influence the outcomes, but these do not seem to have been taken into account or are assumed not to differ between breeds or housing conditions. Again, is there evidence for this assumption?

For aggregate animal welfare, the annual slaughter rate is of crucial importance but not the size of the broiler population at any moment in time. This should be explained better and terminology to differentiate both metrics should be clarified from the very beginning, and these terms should subsequently be used more consistently throughout the manuscript (now terms such population, full population, total population, placements etc... are used and confuse the reader).

For aggregate animal welfare the authors seem to have a one-sided focus on negative welfare outcomes (L99-102, L423-441). L429-432 emphasizes this view and really ought to be better substantiated and references are needed to support this statement. Indeed the longer an animal lives and the more animals that need to be slaughtered, the higher the number of possibly

negative experiences (all other things being kept equal). However, also the number of positive experiences will increase. The authors seem to assume that such positive experiences are much less common as compared to the negative ones, but this ratio might perhaps become positive in case of slower-growing breeds with outdoor access. If so, wouldn't aggregate welfare increase (rather than decrease) the more birds are kept per year and the longer they live? Perhaps this could be elaborated upon.

For the environmental impact, it seems that the feed composition is assumed not to differ depending on breed or housing condition. I'm not a nutritionist, but I'm pretty sure that slower-growing breeds need less protein-rich diets as compared to fast growing breeds. This probably implies that less soy is needed, which I would think has an important impact on the ecological footprint and land-use.

L402-404 I consider animal welfare science to be part of animal science. I don't agree that in animal welfare science only the individual is taken into account. To the contrary, most animal welfare monitoring protocols (e.g. Welfare Quality, AWIN,...) aim to assess the welfare of the group rather than the individual.

===PREPARING YOUR MANUSCRIPT===

If you have been asked to revise the written English in your submission as a condition of publication, you must do so, and you are expected to provide evidence that you have received language editing support. The journal would prefer that you use a professional language editing service and provide a certificate of editing, but a signed letter from a colleague who is a fluent speaker of English is acceptable. Note the journal has arranged a number of discounts for authors using professional language editing services (<https://royalsociety.org/journals/authors/benefits/language-editing/>).

===PREPARING YOUR REVISION IN SCHOLARONE===

To revise your manuscript, log into <https://mc.manuscriptcentral.com/rsos> and enter your Author Centre - this may be accessed by clicking on "Author" in the dark toolbar at the top of the

page (just below the journal name). You will find your manuscript listed under "Manuscripts with Decisions". Under "Actions", click on "Create a Revision".

Author's Response to Decision Letter for (RSOS-210478.R1)

See Appendix C.

RSOS-210478.R2

Review form: Reviewer 3

Is the manuscript scientifically sound in its present form?

No

Are the interpretations and conclusions justified by the results?

Yes

Is the language acceptable?

Yes

Do you have any ethical concerns with this paper?

No

Have you any concerns about statistical analyses in this paper?

No

Recommendation?

Accept with minor revision (please list in comments)

Comments to the Author(s)

I'm happy with most revisions, although I still have some issues.

The authors shouldn't just argue that mortality and rejection rates may change in different directions (L491-499), they ought to refer to the scientific literature or other solid evidence for any such claims. Similarly I would expect the authors to search for evidence of differences in diet composition for fast vs slower growing broilers and discuss the environmental implications accordingly.

I fail to see that the relevant unit for aggregate animal welfare can be anything else apart from the total number of animals that have suffered to a certain degree during their lifetime. The total animal suffering to provide the human population with a certain amount of chicken meat is what matters morally not the total amount of animal suffering on a given day. Thus it still appears to me that aggregate animal welfare is purely based on differences in annual slaughter rate which in turn is almost exclusively determined by differences in slaughter weight between fast vs slower growing breeds in the calculations. This makes it of utmost importance that the slaughter weights that are used in the calculations are realistic, as they are in practice in the US. It should also be pointed out that if slaughter weights between fast vs slower breeds would be more equal (as is the case in Europe), differences in aggregate welfare would also disappear.

I would tend to agree that even a slower growing broiler, during its entire lifespan, would suffer from more negative experiences than enjoy positive emotions, so that the net balance is still negative and the fewer animals that are needed the better for aggregate animal welfare.

However, the net welfare status of the life of an individual slower growing broiler would probably be (much) less negative than that of a fast growing broiler. Is it sufficiently emphasized that this difference ought to be taken into account for the estimation of aggregate welfare? If net welfare status during the life of a slower-growing broiler is 50% higher as compared to that of a fast growing broiler, than aggregate welfare will only be worse if >50% more slower growers are needed as compared to fast growers to provide the same amount of meat to humans...

Decision letter (RSOS-210478.R2)

Dear Professor Chan

On behalf of the Editors, we are pleased to inform you that your Manuscript RSOS-210478.R2 "The "sustainability gap" of US broiler chicken production: trade-offs between welfare, land-use, and consumption" has been accepted for publication in Royal Society Open Science subject to minor revision in accordance with the referees' reports. Please find the referees' comments along with any feedback from the Editors below my signature.

Please submit your revised manuscript and required files (see below) no later than 7 days from today's (ie 25-Apr-2022) date. Note: the ScholarOne system will 'lock' if submission of the revision is attempted 7 or more days after the deadline. If you do not think you will be able to meet this deadline please contact the editorial office immediately.

on behalf of Professor Pete Smith (Subject Editor)
openscience@royalsociety.org

Associate Editor Comments to Author:

Thank you once more for your patience during the review process for your paper, but I hope you'll be reassured that the remaining tweaks are comparatively minor - and once these are addressed in a final revision, the paper will be ready for acceptance. Many thanks for taking seriously the earlier recommendations of the reviewer (to whom we also offer thanks).

Reviewer comments to Author:

Reviewer: 3

Comments to the Author(s)

I'm happy with most revisions, although I still have some issues.

The authors shouldn't just argue that mortality and rejection rates may change in different directions (L491-499), they ought to refer to the scientific literature or other solid evidence for any such claims. Similarly I would expect the authors to search for evidence of differences in diet composition for fast vs slower growing broilers and discuss the environmental implications accordingly.

I fail to see that the relevant unit for aggregate animal welfare can be anything else apart from the total number of animals that have suffered to a certain degree during their lifetime. The total animal suffering to provide the human population with a certain amount of chicken meat is what matters morally not the total amount of animal suffering on a given day. Thus it still appears to me that aggregate animal welfare is purely based on differences in annual slaughter rate which in turn is almost exclusively determined by differences in slaughter weight between fast vs slower growing breeds in the calculations. This makes it of utmost importance that the slaughter weights that are used in the calculations are realistic, as they are in practice in the US. It should also be pointed out that if slaughter weights between fast vs slower breeds would be more equal (as is the case in Europe), differences in aggregate welfare would also disappear.

I would tend to agree that even a slower growing broiler, during its entire lifespan, would suffer from more negative experiences than enjoy positive emotions, so that the net balance is still negative and the fewer animals that are needed the better for aggregate animal welfare. However, the net welfare status of the life of an individual slower growing broiler would probably be (much) less negative than that of a fast growing broiler. Is it sufficiently emphasized that this difference ought to be taken into account for the estimation of aggregate welfare? If net welfare status during the life of a slower-growing broiler is 50% higher as compared to that of a fast growing broiler, then aggregate welfare will only be worse if >50% more slower growers are needed as compared to fast growers to provide the same amount of meat to humans...

===PREPARING YOUR MANUSCRIPT===

one version should clearly identify all the changes that have been made (for instance, in coloured highlight, in bold text, or tracked changes);

===PREPARING YOUR REVISION IN SCHOLARONE===

-- Ensure that your data access statement meets the requirements at <https://royalsociety.org/journals/authors/author-guidelines/#data>. You should ensure that you cite the dataset in your reference list. If you have deposited data etc in the Dryad repository, please only include the 'For publication' link at this stage. You should remove the 'For review' link.

-- If you are requesting an article processing charge waiver, you must select the relevant waiver option (if requesting a discretionary waiver, the form should have been uploaded, see 'File upload' above).

-- If you have uploaded any electronic supplementary (ESM) files, please ensure you follow the guidance at <https://royalsociety.org/journals/authors/author-guidelines/#supplementary-material> to include a suitable title and informative caption. An example of appropriate titling and captioning may be found at https://figshare.com/articles/Table_S2_from_Is_there_a_trade-off_between_peak_performance_and_performance_breadth_across_temperatures_for_aerobic_scope_in_teleost_fishes_/3843624.

Author's Response to Decision Letter for (RSOS-210478.R2)

See Appendix D.

Decision letter (RSOS-210478.R3)

Dear Professor Chan,

I am pleased to inform you that your manuscript entitled "The "sustainability gap" of US broiler chicken production: trade-offs between welfare, land-use, and consumption" is now accepted for publication in Royal Society Open Science.

Please remember to make any data sets or code libraries 'live' prior to publication, and update any links as needed when you receive a proof to check - for instance, from a private 'for review'

URL to a publicly accessible 'for publication' URL. It is good practice to also add data sets, code and other digital materials to your reference list.

Royal Society Open Science is a fully open access journal. A payment may be due before your article is published. Our partner Copyright Clearance Centre will contact the corresponding author about your open access options (if you have any queries regarding fees, please see <https://royalsocietypublishing.org/rsos/charges> or contact authorfees@royalsociety.org).

on behalf of Prof Pete Smith (Subject Editor)
openscience@royalsociety.org

Appendix A

The intention behind this manuscript is sound – there are almost certainly trade-offs between animal welfare and some aspects of sustainability in chicken production systems – and these do require modelling and discussion. However, this paper requires some very substantial revision to be an acceptable and useful part of the debate. The main issues are a lack of industry sector-specific input leading to some very questionable assumptions, a notable lack of understanding of the welfare issues facing broiler chickens (hence the drivers for transition), and the choice to model conventional production against a rather extreme alternative. More details are provided below:

The authors have chosen to model conventional indoor production using a fast-growing breed against perhaps the most extreme alternative (pasture production using one of the slowest-growing breeds). This makes the sustainability “gap” appear far higher than it would be in practice. There are many intermediate breeds which are healthier than Ross 308 but which grow faster than Ranger birds (e.g with growth rates of 40-50g/day and slaughtered at 56 – 60 days, compared with Ranger growth rates of <40g/day).

There are also many intermediate systems that are being proposed that would assure bird welfare but still allow mass production with a far lower sustainability “gap”. Such initiatives include the Better Chicken Commitment which is gaining ground in the USA with companies such as KFC and Burger King signed up. Adherence to these standards would permit indoor production using an intermediate breed. It is FAR more likely that the majority of commercial intensive production would be replaced in this way than by a wholesale shift to pasture. Modelling the impact of switching to an intermediate breed in an intermediate system is ESSENTIAL for the overall credibility of this analysis.

Not all environmental aspects have been considered. The authors mention antibiotic use (line 45) but don't mention that antibiotic use could be significantly reduced if healthier birds were housed at lower stocking densities. The authors therefore need to set out far more clearly which aspects of sustainability they are, and are not, measuring. Similarly, the authors mention that mortality in conventional systems is around 5%. Using healthier breeds would reduce mortality (line 349-350) but the effect of this has not been modelled. These are positive counter-influences to the higher environmental resource costs that have been included. They must be considered.

It is also the case that any pasture or outdoor range would not be used for the full period of a bird's life. The first few weeks would be spent indoors until chicks were old enough to be let out. Land could potentially be rested or used in other ways. Many organic systems have shared land use with trees, bushes or other crops on the same area that chickens occupy. This is another counter (or at least mitigating) influence.

Previous research looking at sustainability (different aspects) and welfare trade-offs has not been mentioned e.g. Leinonen et al., 2014, Poultry Science; Tallentire et al., 2019. International Journal of Life Cycle Assessment; van Wagenburg et al., 2017, Animal.

There is a difference between stocking rate (birds/m² or birds/square foot) and stocking density, which for broilers is always expressed as weight/are (e.g. kg/m²). High stocking densities only occur towards the end of production as birds become heavy. Stocking rates will remain the same. Often high stocking densities are managed by thinning (removal of part of the flock for slaughter some days before the rest, thus keeping stocking densities below upper guidelines. This is an important practice not mentioned or considered. In this paper, stocking rate and stocking density are not differentiated.

There is some discussion of the possibility of raising Ross 308 birds on pasture – this would never happen in commercial practice, as birds of this breed are not at all robust and mortality would be extreme. It is not clear why this option is modelled as it would not happen.

Many of the welfare issues are mentioned that do not apply to broiler chickens at all. These include beak trimming (line 44) which applies to laying hens and some broiler breeders (broilers reared for meat production are not beak trimmed); feather pecking and cannibalism (line 46) (broilers do not show these behaviours and hence lines 326-228 is nonsensical); bone breakage (line 63 – bone breakage is an issue for laying hens not broilers – the reference cited does not mention bone breakage and so this whole section is completely incorrect).

In contrast, the major welfare issues facing broiler chickens are barely mentioned or not mentioned at all. The major welfare challenges for broiler chickens include contact dermatitis (not mentioned), lameness or leg disorders (not mentioned, except with passing and unexplained mention of gait score on line 68-70), ascites (not mentioned) and chronic hunger for broiler breeders (not mentioned). Since most of the welfare issues mentioned do not apply to broilers, and the welfare issues *not* mentioned are the critical drivers behind the call for slower-growing breeds, it is hard to trust that the paper is knowledgeable about any other aspects of broiler production.

A conception or definition of welfare is also not given and phrases such as “the welfare of the whole population” (line 54) make no sense. Populations are composed of individual animals, so if individual animal welfare is improved the welfare of the population will also be improved. What sort of trade-offs do the authors envisage that could lead to a trade-off between individual and population welfare in the case of broilers?

In Europe broilers are never confined in cages. In the USA broilers are rarely confined in cages. Broilers are generally raised in unconfined floor systems in flock sizes up to many tens of thousands of birds. Given this, it is very strange to rely on an experimental study of caged broilers to make general points about stocking density (lines 77 – 80). A high stocking density in a cage may well result in bruising and feather loss, but this is not applicable to the general USA broiler population. In floor systems, high stocking densities may lower litter condition and increase mortality or the risk of skin dermatitis but this is not made clear. Again, this displays a limited and incorrect view of the industry that is being modelled.

Line 76 – states that slower-growing birds spend more time in “confinement” but this is not true. First, these birds are not caged (see above) and broilers are rarely confined in any meaningful way. Towards the end of life when stocking densities are very high birds may be restricted in their movements by other birds – but for the slower growing birds this would only occur at the very end of life and not for any longer than the restriction faced by faster-growing birds at an earlier age.

As a general point, for a journal intended for an international audience, I would advise presenting all figures using current measurement systems e.g. kg/m² rather than lbs/square foot. Where USA industry persists in using older measures then these can be given alongside the conversion.

Appendix B

We wish to thank the reviewers for their extensive and helpful feedback. The paper has been greatly improved by incorporating their suggestions and we are very grateful for the considerable time and thought that they devoted to this work.

Below we provide a point-by-point response to each item identified by the reviews. Our responses are in blue font, the reviewers' original comments are in black font.

Reviewer: 1

The intention behind this manuscript is sound – there are almost certainly trade-offs between animal welfare and some aspects of sustainability in chicken production systems – and these do require modelling and discussion. However, this paper requires some very substantial revision to be an acceptable and useful part of the debate. The main issues are a lack of industry sector-specific input leading to some very questionable assumptions, a notable lack of understanding of the welfare issues facing broiler chickens (hence the drivers for transition), and the choice to model conventional production against a rather extreme alternative. More details are provided below:

Thank you for this positive assessment of the importance of our research area and for your constructive ideas for improving our work. We made substantial revisions to this analysis, following reviewer 1's suggestions. The most substantial additions include:

- A. Incorporating an analysis of two additional broiler breeds with “intermediate” growth rates: the Ranger Gold and Ranger Classic, using industry data. Our updated results reflect a wider range of “sustainability gaps” reflecting more numerous options for higher-welfare breeds.
- B. Included a management scenario of partial outdoor access in accordance with GAP minimum standards
- C. Corrections in our introduction and discussion, discussing welfare-related challenges as those related to specifically to broiler chickens.

We thank the reviewer for suggesting these modifications. We believe their inclusion has strengthened the rigor and breadth of our analysis. Further details and line-by-line response are below

The authors have chosen to model conventional indoor production using a fast-growing breed against perhaps the most extreme alternative (pasture production using one of the slowest-growing breeds). This makes the sustainability “gap” appear far higher than it would be in practice. There are many intermediate breeds which are healthier than Ross 308 but which grow faster than Ranger birds (e.g with growth rates of 40-50g/day and slaughtered at 56 – 60 days, compared with Ranger growth rates of <40g/day).

We agree that this scenario was the most “extreme” and have thus included intermediate breeds (more above and below). Furthermore, we loved the principal of a “sustainability gap” and, with the reviewer's permission, we would like to use this pithy comment in our revised title!

There are also many intermediate systems that are being proposed that would assure bird welfare but still allow mass production with a far lower sustainability “gap”. Such initiatives include the Better Chicken Commitment which is gaining ground in the USA with companies such as KFC and Burger King signed up. Adherence to these standards would permit indoor production using an intermediate breed. It is FAR more likely that the majority of commercial intensive production would be replaced in this way than by a wholesale shift to pasture. Modelling the impact of switching to an intermediate breed in an intermediate system is ESSENTIAL for the overall credibility of this analysis.

Including intermediate options in our analysis is an excellent idea. We have now implemented this idea to show a range of options available to production, including analyses for two additional breeds and one intermediate housing system (with outdoor access) as recommended. These new analyses have led to substantial changes throughout the MS (from abstract through to discussion). Please see updated MS for all the attendant changes and thank you for this valuable suggestion.

Not all environmental aspects have been considered. The authors mention antibiotic use (line 45) but don't mention that antibiotic use could be significantly reduced if healthier birds were housed at lower stocking densities. The authors therefore need to set out far more clearly which aspects of sustainability they are, and are not, measuring.

We agree that our research focused on land use impacts and did not set out to provide an exhaustive review of all types of environmental impact. We have clarified this focus by updating the title, the abstract, and changing our language where necessary throughout the manuscript, clarifying at all stages that we are quantifying land use, but that this metric is pertinent for a swath of related environmental impacts.

Thank you for also pointing out the possibility that antibiotic use could be decreased in production systems with lower stocking densities, an important point that we now mention in the introduction.

Similarly, the authors mention that mortality in conventional systems is around 5%. Using healthier breeds would reduce mortality (line 349-350) but the effect of this has not been modelled. These are positive counter-influences to the higher environmental resource costs that have been included. They must be considered.

This is an important consideration that we now include in the discussion. Generating a precise change in mortality for all breeds and under all conditions now under consideration in the MS would be impossible, however, and to attempt to do so would therefore introduce speculations into the estimates. Therefore we believe the analyses should proceed under the currently implemented estimates, which do not involve hypothetical mortality rates. In our discussion, we have therefore included the possibility of reduced mortality in the healthier breeds as a potential moderator to our findings and needed area for future research, indicating the need for future analysis.

It is also the case that any pasture or outdoor range would not be used for the full period of a bird's life. The first few weeks would be spent indoors until chicks were old enough to be let out. Land could potentially be rested or used in other ways. Many organic systems have shared land use with trees, bushes or other crops on the same area that chickens occupy. This is another counter (or at least mitigating) influence.

This is another interesting consideration for more sustainable production. In our discussion, we now state that a hybridized method of partial or rotating pasture access would likely land in between the two scenarios that we quantified (partial outdoor access and full-time pasture management). Shared land use with trees, bushes and crops is compelling, but it is unclear how scenarios such as this would differ from land use scenarios wherein pasture was maximally intensively used to conserve more space for native vegetation and crops elsewhere. This taps into a much larger debate of land sparing vs. land sharing that we believe lies outside the scope of our analysis.

Previous research looking at sustainability (different aspects) and welfare trade-offs has not been mentioned e.g. Leinonen et al., 2014, Poultry Science; Tallentire et al., 2019. International Journal of Life Cycle Assessment; van Wagenburg et al., 2017, Animal.

We now reference these studies in the introduction and discuss their relevance to our introduction, while framing our novel contribution in a more measured way.

There is a difference between stocking rate (birds/m² or birds/square foot) and stocking density, which for broilers is always expressed as weight/are (e.g. kg/m²). High stocking densities only occur towards the end of production as birds become heavy. Stocking rates will remain the same. Often high stocking densities are managed by thinning (removal of part of the flock for slaughter some days before the rest, thus keeping stocking densities below upper guidelines. This is an important practice not mentioned or considered. In this paper, stocking rate and stocking density are not differentiated.

Thank you for alerting us to the need to be more precise in the description of our methods—we now clarify that we refer to stocking rates (birds per unit area) in our introduction and methods.

There is some discussion of the possibility of raising Ross 308 birds on pasture – this would never happen in commercial practice, as birds of this breed are not at all robust and mortality would be extreme. It is not clear why this option is modelled as it would not happen.

We understand that raising Ross 308 on pasture is currently unlikely to happen at full-scale commercial practice. That said, there are some small-scale farms that do report farming Ross on pasture (e.g. <https://www.sare.org/wp-content/uploads/Profitable-Poultry.pdf>) so there is a potential of scaling up these practices, though the degree to which it can be scaled remains unknown. In principle, at least, farming Ross on pasture could occur at a greater scope than is presently seen. For completeness sake, this data point is necessary for our ability to weigh all possibilities, regardless of how remote they seem. However, we do appreciate the reviewer's point that the profiles of this breed complicate raising them on pasture.

Many of the welfare issues are mentioned that do not apply to broiler chickens at all. These include beak trimming (line 44) which applies to laying hens and some broiler breeders (broilers reared for meat production are not beak trimmed); feather pecking and cannibalism (line 46) (broilers do not show these behaviours and hence lines 326-228 is nonsensical); bone breakage (line 63 – bone breakage is an issue for laying hens not broilers – the reference cited does not mention bone breakage and so this whole section is completely incorrect).

In contrast, the major welfare issues facing broiler chickens are barely mentioned or not mentioned at all. The major welfare challenges for broiler chickens include contact dermatitis (not mentioned), lameness or leg disorders (not mentioned, except with passing and unexplained mention of gait score on line 68-70), ascites (not mentioned) and chronic hunger for broiler breeders (not mentioned). Since most of the welfare issues mentioned do not apply to broilers, and the welfare issues not mentioned are the critical drivers behind the call for slower-growing breeds, it is hard to trust that the paper is knowledgeable about any other aspects of broiler production.

We agree that we have mistakenly misconstrued the welfare issues facing broiler chickens and understand that this is a concerning oversight. We thank the reviewer for pointing it out to us and for providing valuable corrective information. We understand that welfare risks are very different for different breeds. As the reviewer suggested, it is preferable to focus this section on the breeds and systems at hand and have updated the relevant passages in our introduction and discussion accordingly, which includes specific references to the welfare challenges that the reviewer identifies.

A conception or definition of welfare is also not given and phrases such as “the welfare of the whole population” (line 54) make no sense. Populations are composed of individual animals, so if individual animal welfare is improved the welfare of the population will also be improved. What sort of trade-offs do the authors envisage that could lead to a trade-off between individual and population welfare in the case of broilers?

We have now included more detailed explanation for our motivation of analyzing chicken population and slaughter rate outcomes for shifts to slower-growing breeds. Increasingly welfare researchers are focusing not just on average welfare conditions for each bird, but the so called “aggregate” welfare of entire populations. As a crude formulation, if an intervention were to make birds’ lives 50% less negative, but increased the population by 10 times, then the aggregated welfare overall would still be worse by a factor of 5. We describe in our introduction that this level of aggregation is common in public health spheres (e.g. DALYs and QALYs) but is only more recently embraced in animal welfare research, with relevant references included in the introduction.

Throughout the introduction and discussion, we now refer to this type of determination as “aggregated” welfare rather than population-level welfare, and clarify that this analysis provides a building block for determinations of aggregated welfare in the future. Furthermore, in our discussion, we state that despite ambiguity and ongoing research regarding the most appropriate methods for aggregating welfare, our analysis demonstrates that the consequences for aggregated welfare in shifting to better breeds are not clear-cut because populations will increase, therefore shifting to a slower-growing breed involves tradeoffs.

In Europe broilers are never confined in cages. In the USA broilers are rarely confined in cages. Broilers are generally raised in unconfined floor systems in flock sizes up to many tens of thousands of birds. Given this, it is very strange to rely on an experimental study of caged broilers to make general points about stocking density (lines 77 – 80). A high stocking density in a cage may well result in bruising and feather loss, but this is not applicable to the general USA broiler population. In floor systems, high stocking densities may lower litter condition and increase mortality or the risk of skin dermatitis but this is not made clear. Again, this displays a limited and incorrect view of the industry that is being modelled.

Line 76 – states that slower-growing birds spend more time in “confinement” but this is not true.

First, these birds are not caged (see above) and broilers are rarely confined in any meaningful way.

Towards the end of life when stocking densities are very high birds may be restricted in their movements by other birds – but for the slower growing birds this would only occur at the very end of life and not for any longer than the restriction faced by faster-growing birds at an earlier age.

We thank the reviewer for their feedback. We agree that US broilers are rarely confined in cages. Pertaining to the previous comment, we now discuss welfare issues related to stocking densities and housing systems that are unique to broilers. In a few key areas, we discuss issues of such stocking densities, for which we now use “crowding” as a shorthand, and detail that these issues unique to broilers include jostling, climbing, and waking each other from rest.

As a general point, for a journal intended for an international audience, I would advise presenting all figures using current measurement systems e.g. kg/m² rather than lbs/square foot. Where USA industry persists in using older measures then these can be given alongside the conversion.

Per the reviewer’s concern and suggestion, we have now shifted to the metric system, with the exception of a few input parameters in our methods, which we convert later to produce results that are entirely in metric units.

Reviewer: 2

It is the overall impression that the manuscript contains interesting information since welfare and environmental issues are important in any animal production. Since it is an American study, the legislation regarding production of conventional broiler chickens is probably different from the European, so I find it important that there are some information about the American legislation given in the text. Further, I find part of the Introduction very subjective in the description of the conventional broiler production, which probably gives a skewed view of the production compared to alternative productions. More concrete information should be given to explain in an objective way the conventional and alternative broiler productions in the United States.

Thank you for your positive assessment of the general scope of our research. We have extensively revised the MS in response to your comments and the comments from the other reviewer. In doing so, we have included more precise and concrete descriptions of US broiler production systems. In

addition, in our discussion we now clarify relevant legislation in the United States and their enforcement by executive agencies in the USDA, including the Humane Methods of Slaughter Act and the so called 28-Hour law for farmed animal transportation. These modifications have greatly improved the MS and we are grateful to the reviewer for their constructive feedback. Please see below for detailed responses to all comments.

Introduction:

It is important to make it clear in the text that the paper is dealing with broiler chickens for meat production and not chickens (pullets) for egg production and if it is the conventional production or an alternative production. Therefore, it should be written already in the first line Page 3, 38: In 2018, the conventional broiler chicken production (delete industry) -----.

Important distinction—thank you! We have now clarified as follows:

In 2018, the conventional broiler chicken production generated over 31.7 billion US dollars, with over nine billion chickens slaughtered in the United States¹⁵, and the overall consumption of chicken meat in the U.S. continues to rise¹⁶. Broiler chickens are raised for meat, which are different breeds than chickens who lay eggs.

In line 43 delete known as broilers.

Deleted.

Line 40: suggests: as the demand for chicken meat production increases, so too---: what does this mean? It is not very clear: who have concerns – the consumers?

We have clarified as follows:

As the demand for chickens increases, concerns also arise regarding the individual welfare of the chickens in these systems and the damage such practices cause to the surrounding ecosystems and environment

Line 42: after environment, such as (list some few examples).

We have added the following examples: greenhouse gas emissions, freshwater pollution, and encroachment of feed cropland into native ecosystems

Line 44-45: by mentioning these practices, the conventional broiler production appear very negative and is it really correct that broiler chickens are beak trimmed? Due to their very short life span, this is not normal practise (at least in Europe) and therefore an unnecessary cost for the farmer. Feather pecking is mainly seen with some layer genotypes having a long production cycles. The reference Nicol, C (2018) is dealing with egg layers and not broilers and it also appear to be the case for the reference by Schwean-Larden, K. It is suggested that the authors find more relevant references, and inform about the practice for broilers and not layers, since these productions are very different. This is actually mentioned by the authors on page 7, line 120-121.

Thank you for raising these issues. We originally intended to provide chicken welfare issues in general to discuss trade-offs generically, but now realize that our intention was not clear and that, in any case, it would be preferable to focus on broiler-specific issues. We have updated this passage in our introduction accordingly.

Stocking densities: what are the legislation for stocking densities? For instance in kg/pounds per m² at slaughter age. If there is no legislation in this area, it should also be mentioned.

There are no legal protections for farmed animals anywhere in the US on the farms. The only legislation concerning their handling and management pertain to transport and slaughter, and poultry aren't even covered in this legislation because they have not legally been defined as "livestock" by regulating US agencies. We now include this in the discussion.

Line 46: cannibalism – the references are related to pullets/layers and not broilers. Cannibalism is not seen very often in broiler production, and at least references should deal with cannibalism in broilers.

Thank you for pointing out our lack of specificity here. In updating the broiler-specific welfare issues, we have also removed this reference to cannibalism.

Page 4: in published papers, presentation of a topic should be objective and academic, and therefore words, such as cruelty, should be avoided, or if it is a statement from e.g. an animal welfare organisation, it could be given as a citation "----" and with a reference included. Line 49-50. Suggest to change the text to: --- alternative ways to produce broiler chickens from the conventional, intensive system ----.

We have updated the text as suggested.

Line 60-62: is it really common practice to have a slaughter age at 47 dage??? Ross 308 is very fast growing and continuing the production until 47 age will be very expensive in feed costs and the chickens will probably not be able to walk very well the last 1-2 weeks, which increase the risk for wounds and breast blisters, since the chickens will sit down on the litter more often. This could result in a higher discarding percentage at the slaughter house, which reduce the payment to the farmer. It there any information about these data available? If some kind of feed restriction is used to reduce weight gain, it should be mentioned. In the recent Ross 308 management guide (2021) from Aviagen the final live weight at 47 days is 3,75kg (8,33pounds), which are in line with the data given on market weight, so probably, the chickens have been feed ad. Lib.

We agree with the reviewer, so took the following steps to normalize the data. We took the USDA live weight and average age and scaled this from the Aviagen data to normalize the weight for the other three breeds we analyzed.

<https://ask.usda.gov/s/article/How-old-are-chickens-used-for-meat>

<https://www.nationalchickencouncil.org/statistic/us-broiler-performance/>

As we discuss in our methods, data from the companies should reflect ideal conditions, which can be less. From our demographic model, which includes population, placement rates, and slaughter,

we can deduce that the average turnover rate or duration that Ross chickens on conventional farms is 47 days, somewhat higher than the Aviagen-quoted 42 day period. There are a number of areas wherein we can see that industry-provided data are slightly more optimistic than in the real world, which comports with their business interests.

Line 66: higher instead of greater

Corrected.

Line 69-71. Some of the parameters mentions are repetitions from line 67-69.

Corrected.

Page 6, line 95: regarding health consequences, references should be found dealing with e.g. risk of coccidiosis or at least mention the importance of different vaccinations. The ref. Chen et al (2013) is more dealing with the effect of outdoor area on meat quality and not about potential risk for pathogen infections. This information should be added to the text.

We have scaled back our claim to state that there are “few” health consequences (not none), removed the Chen reference in this line, and added a recent reference to address risk of coccidiosis in pastured chickens.

Line 99: write nutrient requirement instead of feed requirement.

Updated as suggested

Line 105: add conventional broiler before chickens

Phrase “conventional broiler” added.

Methods

This paragraph includes a lot of equations for calculating the different results needed to discuss the topic and the explanation given is overall sufficient and understandable.

Thank you for this positive feedback!

Line 164-165: what is the difference between the current conventional system and CAFO's? CAFO's are defined on page 3, line 43, but it is not clear how the current conventional system differs?

We now clarify that “the current conventional system” refers to Ross 308 broiler chickens raised in CAFOs.

Line 168: it is probably 0.0101 km².

Corrected.

Why not give the data also in m² per chicken?

We have added data regarding m² per chicken.

Line 172 ----: a general comment: it should be mentioned somewhere that the alternative, outdoor systems are free-range, but not organic?? Production of organic feed would give another dimension to the calculations, which is probably not the case in this study, but it should be mentioned that the alternative systems considered are based on conventional feed ingredients.

This is a good point. Now clarified as follows:

Part of Global Animal Partnership's certification includes outdoor access, which is not the same as organically raised chicken. Some farms also raise chickens on pasture and are also not the same as organically raised chickens.

The use of the outdoor area by the broilers is very much dependant on the vegetation (trees and bushes/ or not) and the genotypes. Is this considered or is it "defined" that the broilers will use the total area regardless of the extent of planting?

The conditional regarding the presence vs. absence of vegetation is important. We now clarify in our discussion that we assume that pasture is entirely "open" and make no specific assumptions regarding natural large vegetation such as trees or bushes, which would provide shelter for chickens, habitat for grubs, and ecosystem services, but could increase land requirements further if such native vegetation is too dense. We clarify that this is an area where further land use research is needed.

Results: the description and explanation of the results appear sufficient, however, with a lot of information placed in three figures and one table. It is helpful that the equations used for the different calculations are included in brackets in each paragraph.

We have added references to the relevant equations to paragraphs as suggested.

Discussion.

Page 16, line 314-315: How to define animal welfarists?? There are many animal scientist that have focussed on how to improve welfare for animals in different systems, Scientists working with nutrition also look into alternative feeding strategies and the use of new protein sources for a more sustainable production, so this "statement"/sentence should be reformulated. So, scientists working with nutrition and performance and scientist working with welfare issues. Like the Introduction, it is important to be objective.

We agree that more clarity and nuanced language is required here and have updated the passage accordingly:

It is our intention that these results highlight opportunities for scientists working with nutrition and performance, scientists working with welfare issues, and animal

welfare organisations to work in tandem, to tackle important trade-offs between welfare and land usage and its attendant impacts

Line: it is very good to introduce the One Health/Welfare approach, but the cooperation between nutrition/performance and welfare disciplines are not new.

We agree, and have modified the discussion to indicate that our analysis add to this integration by connecting performance with welfare, population, and environment throughout this section.

Page 17: line 339-341: Nonetheless, ----, for what reasons? It is not clearly explained? If the broilers have sufficient outdoor space, where they can express their natural behaviour, what factors will results in more stress and mortality: predation? This situation will probably be dependant of the quality of the outdoor area with regard to the amount of vegetation and areas with trees, where the broilers can seek protection against raptors. This is described more in details in the following lines 342-352, which is relevant information, but the lines 339-341, still appear confusing.

We agree and we have removed this sentence. Elsewhere we clarify that welfare concerns remain because of the potential for predation, as well as well-document distress caused to a higher population of birds throughout transport and slaughter processes, which are unregulated in the US (with citations included).

The discussion from line 361 --- regarding the needed change in how humans consume meat is very important on order to deal with the environmental and welfare issues for the animals, bur also regarding health and welfare for humans.

We would like to include this point, but the evidence regarding the role of chickens in human health is complicated, with experts disagreeing on the role of poultry meat in nutrition among diets in developed countries including the US. Elsewhere, we discuss concerns regarding the emergence of antibiotic resistant pathogens. We thank the reviewer and hope the larger set of improvements that we have made to the discussion following their comments ultimately broadens the impacts of our analysis.

Appendix C

We wish to thank the reviewers for their extensive and helpful feedback. The paper has been greatly improved by incorporating their suggestions and we are very grateful for the considerable time and thought that they devoted to this work.

Below we provide a point-by-point response to each item identified by the reviews. Our responses are in blue font, the reviewers' original comments are in black font.

Reviewer comments to Author:

Reviewer: 2

Comments to the Author(s)

In this corrected version of the manuscript (Manuscript ID RSOS-210478.R1), you have followed most of the recommendations given for improvement of the paper and in the cases, where you haven't agreed on a suggestion for changes, there are given good arguments for not including the correction: example: Reviewer: The discussion from line 361 --- regarding the needed change in how humans consume meat is very important on order to deal with the environmental and welfare issues for the animals, but also regarding health and welfare for humans. You have given a justified explanation for NOT including this aspect. The corrected paper has been improved to a large extent and when corrections are accepted in a final version of the manuscript, a thorough evaluation of the text has to be carried out for spelling errors and for a more fluent language, before publication, if finally accepted.

We thank the reviewer for the positive assessment of our revised MS. The reviewer's comments have helped us greatly improve the manuscript.

Reviewer: 3

Comments to the Author(s)

This is a very interesting and thought-provoking study, with potentially far-reaching implications. These consequences on the annual slaughter rate differences between breeds are almost exclusively caused by differences in slaughter weight between fast and slower growing breeds. This needs emphasizing because in other parts of the world, I think, these differences in slaughter weight can be very different. I'm from Europe, and Ross 308 tend to be slaughtered at a much lower weight and I think slower growing breeds reach more or less the same slaughter weight as the Ross 308. I presume that the authors have thoroughly checked the slaughter weights they have used for their calculations. In any case they ought to limit their conclusions to parts of the world where differences in slaughter weight are similar and perhaps point out that this may be very different in other parts of the world.

We thank the reviewer for this positive assessment of the importance of our research area and for the constructive ideas for improving our work.

We now discuss more explicitly in our methods that performance data from companies may reflect ideal conditions. In the real world, the growth and FCRs can be less due to variable climate, pathogens, and feed quality. Instead, we used a top-down demographic model constrained by USDA data, which includes populations, placement rates, and slaughter, to deduce that the average turnover rate or duration that Ross chickens on conventional farms is 47 days, somewhat higher than the Aviagen-quoted 42 day period. There are a number of areas wherein we can see that industry-provided data are slightly more optimistic than in the real world, which comports with their business interests.

We have clarified potential confusion in the manuscript by emphasizing that the data analysed in this study is based on the average growth statistics for broilers raised in the US only and does not reflect other parts of the world.

If slower-growing breeds would be slaughtered at the same weight, than aggregate welfare would not or hardly been affected according to this study. It surprises me that in these calculations the dressing percentage seems to have been kept constant between breeds (and housing conditions). Is there evidence to support that dressing percentage is indeed not affected by breed nor by outdoor access?

We thank the reviewer for their feedback. We have updated the dressing percentages for each breed according to the information provided by Aviagen for all four breeds. We have updated the results throughout the manuscript. Because dressing percentages for the slower-growing breeds are typically lower than the Ross broilers, this results in increased resource requirements and slaughter rates slightly for all three slower-growing breeds.

Apart from differences in mortality (which aren't taken into account and "dealt with" a bit easily in the Discussion, L464-467), differences in carcass rejection rates (at slaughter) may also influence the outcomes, but these do not seem to have been taken into account or are assumed not to differ between breeds or housing conditions. Again, is there evidence for this assumption?

The reviewer is correct that we did not model differences in carcass rejection rate. Like mortality, carcass rejection rate could be reasoned to go up *or* down in the alternative production systems, but to our knowledge, there is no consensus indicating which direction is most likely. We now include a list of considerations regarding mortality and carcass rate which warrant further analysis, adding more substance to this discussion in the section that the reviewer indicated. Hopefully this analysis can motivate further necessary research so that these considerations can be accounted for and modeled in the future.

For aggregate animal welfare, the annual slaughter rate is of crucial importance but not the size of the broiler population at any moment in time. This should be explained better and terminology to differentiate both metrics should be clarified from the very beginning, and these terms should subsequently be used more consistently throughout the manuscript (now terms such population, full population, total population, placements etc... are used and confuse the reader).

The reviewer is correct that the annual slaughter rate is generally understood to be of more relevance to aggregate animal welfare than the size of the broiler population at any moment in

time. This emphasis on annual slaughter rate is consistent with considering the individual animal as the unit of analysis. Considering time as the unit of analysis, however, population becomes centrally relevant to discussions about welfare. For example, on any given day in the U.S., conditions that involve higher chicken populations would amount to worse welfare in the aggregate than conditions that involve lower chicken populations (all else being equal and assuming a negative welfare state—see next comment for more details on our rationale for this assumption). In other words, while the dynamics at play when evaluating welfare in the aggregate are invariably complex, population is one of several relevant parameters to consider. We have updated the manuscript's introduction to clarify these points.

We also agree that greater consistency in our terminology (with clear definitions at the outset) would aid readability and clarity of our results.

As such, in response to this helpful comment, we have (i) defined 'aggregate welfare' at the country-level scale as distinct from the 'population', which now refers only to the total number of broiler chickens in existence in the US at given point in time, and updated the remainder of the MS to ensure that we use the same terms throughout the paper, and (ii) clarified the role that annual slaughter rate plays in animal welfare.

For aggregate animal welfare the authors seem to have a one-sided focus on negative welfare outcomes (L99-102, L423-441). L429-432 emphasizes this view and really ought to be better substantiated and references are needed to support this statement. Indeed the longer an animal lives and the more animals that need to be slaughtered, the higher the number of possibly negative experiences (all other things being kept equal). However, also the number of positive experiences will increase. The authors seem to assume that such positive experiences are much less common as compared to the negative ones, but this ratio might perhaps become positive in case of slower-growing breeds with outdoor access. If so, wouldn't aggregate welfare increase (rather than decrease) the more birds are kept per year and the longer they live? Perhaps this could be elaborated upon.

The reviewer raises interesting and important questions. These points deserve further consideration than we provided in the original MS and we have updated the relevant passages accordingly.

It is true that our aggregation projections focused on the negative aspects of chicken welfare. Under the current production practices and legal systems (especially in the US), a net-negative intra-individual aggregation seems like a reasonable assumption. Nonetheless, we have added content to both the introduction and discussion of why we make a conditional assumption, while identifying the scope for further nuance through research and debate. The reviewer is correct that under certain conditions, increased positive experiences could outweigh the negative aspects and that at this tipping point, more lives wouldn't automatically translate to worse welfare at scale. Interestingly, it is under those precise conditions, however, that the aggregate welfare costs of slaughter increase because if the life is net-positive, premature death robs the individual of additional positive quality-adjusted life years. While the relative importance of the premature death issue may be debatable, we raise it to illustrate the complications with even intra-

individual aggregations of welfare (see Sandoe et al., 2019, cited in the revised manuscript introduction and discussion).

Because disentangling the nuances and details of such aggregation debates is beyond the scope of the present work, we have instead opted to remain mostly under the assumption that any production systems currently under consideration are likely to lean towards the negative end of the scale. To make this decision more transparent and substantiated, we have (1) adjusted our language in both passages indicated by the reviewer to be more transparent about our reasoning and (2) removed the sentence flagged by the reviewer as the most extreme articulation of this position, and (3) added qualifying text to both the introduction and discussion of the positive-negative welfare ratio issue.

For the environmental impact, it seems that the feed composition is assumed not to differ depending on breed or housing condition. I'm not a nutritionist, but I'm pretty sure that slower-growing breeds need less protein-rich diets as compared to fast growing breeds. This probably implies that less soy is needed, which I would think has an important impact on the ecological footprint and land-use.

Thank you for pointing this out. The diet of slower-growing breeds is important on the ecological footprint and land-use. In our discussion, we now raise this concern alongside a host of other nutritional differences between breeds and management conditions that warrant further analysis and are unfortunately outside the scope of our analysis. We clarify that more representative and distinct diets of each breed under each housing condition is an area where further research is needed.

L402-404 I consider animal welfare science to be part of animal science. I don't agree that in animal welfare science only the individual is taken into account. To the contrary, most animal welfare monitoring protocols (e.g. Welfare Quality, AWIN,...) aim to assess the welfare of the group rather than the individual.

This sentence is unnecessary for the flow of the discussion and based on the reviewer's feedback, we have opted to delete it to avoid confusion.

Appendix D

We wish to thank the reviewers for their extensive and helpful feedback. The paper has been greatly improved by incorporating their suggestions and we are very grateful for the considerable time and thought that they devoted to this work.

Below we provide a point-by-point response to each item identified by the reviews. Our responses are in blue font, the reviewers' original comments are in black font.

The authors shouldn't just argue that mortality and rejection rates may change in different directions (L491-499), they ought to refer to the scientific literature or other solid evidence for any such claims. Similarly I would expect the authors to search for evidence of differences in diet composition for fast vs slower growing broilers and discuss the environmental implications accordingly.

Unfortunately, scientific literature on the relative mortality and rejection rates under the conditions considered in this study are not available. In the absence of such information, the potential directionality and magnitude of the effects could be reasoned to trend in a variety of ways and in our assessment, all possibilities remain plausible (increase, decrease, no change). To ensure readers understand that we are not making strong claims in this passage and are instead, simply discussing the full range of logical possibilities, we have amended the passage to clarify that our consideration of different directions was just that—speculation about plausible pathways. We have also further clarified that the need for research is, therefore, critical. (see L455 – 460 in tracked-changes MS).

I fail to see that the relevant unit for aggregate animal welfare can be anything else apart from the total number of animals that have suffered to a certain degree during their lifetime. The total animal suffering to provide the human population with a certain amount of chicken meat is what matters morally not the total amount of animal suffering on a given day. Thus it still appears to me that aggregate animal welfare is purely based on differences in annual slaughter rate which in turn is almost exclusively determined by differences in slaughter weight between fast vs slower growing breeds in the calculations. This makes it of utmost importance that the slaughter weights that are used in the calculations are realistic, as they are in practice in the US. It should also be pointed out that if slaughter weights between fast vs slower breeds would be more equal (as is the case in Europe), differences in aggregate welfare would also disappear.

We agree with the reviewer that this point is of the utmost importance and we understand that the reviewer remains unconvinced by our position. Nonetheless, “total animal suffering” is the bedrock issue at stake and on this front, we are in complete alignment with the reviewer.

Total animal suffering involves the number of lives affected (as indicated by the reviewer), but it also involves the duration of the suffering: all else being equal, longer durations of suffering equates to worse welfare than shorter durations of suffering. In other words, two days of suffering is worse than one day of suffering.

Unlike annual slaughter rate, population metrics capture the durational aspect of total welfare. As such, even if the annual slaughter rate was the same for both breeds, greater populations of slow growing breeds would, by definition, involve more days of suffering than fast growing breeds. We have further amended the introduction to make this durational point more clear (see L103-106 in tracked-changes MS).

I would tend to agree that even a slower growing broiler, during its entire lifespan, would suffer from more negative experiences than enjoy positive emotions, so that the net balance is still negative and the fewer animals that are needed the better for aggregate animal welfare. However, the net welfare status of the life of an individual slower growing broiler would probably be (much) less negative than that of a fast growing broiler. Is it sufficiently emphasized that this difference ought to be taken into account for the estimation of aggregate welfare? If net welfare status during the life of a slower-growing broiler is 50% higher as compared to that of a fast growing broiler, then aggregate welfare will only be worse if >50% more slower growers are needed as compared to fast growers to provide the same amount of meat to humans...

We very much appreciate the reviewer's agreement that in general, net welfare balance is likely negative, regardless of breed. While the specific differential in overall welfare by breed could be 50%, it seems plausible that it could also be much closer and, crucially, conditional on the farming and/or environmental context. The precise nature of the differential is an important area for further research and the need to determine the boundary conditions of such estimates is made clear by our present findings. At this stage, however, we believe that further speculation on this front is beyond the scope of the present work.